# Insular cortical circuits as an executive gateway to decipher threat or extinction memory via distinct subcortical pathways

Qi Wang[1,2,8], Jia-Jie Zhu[2,8], Lizhao Wang[2], Yan-Peng Kan[2], Yan-Mei Liu[1,2], Yan-Jiao Wu[1,2], Xue Gu[1,2], Xin Yi[1,2,3], Ze-Jie Lin[1,2], Qin Wang[2], Jian-Fei Lu[1,2], Qin Jiang[1,2], Ying Li[1,2], Ming-Gang Liu[1,2], Nan-Jie Xu[2], Michael X. Zhu[4], Lu-Yang Wang[5,6], Siyu Zhang[1,2,7] ✉, Wei-Guang Li[1,2,3,7] ✉ & Tian-Le Xu[1,2,7] ✉

Threat and extinction memories are crucial for organisms' survival in changing environments. These memories are believed to be encoded by separate ensembles of neurons in the brain, but their whereabouts remain elusive. Using an auditory fear-conditioning and extinction paradigm in male mice, here we discovered that two distinct projection neuron subpopulations in physical proximity within the insular cortex (IC), targeting the central amygdala (CeA) and nucleus accumbens (NAc), respectively, to encode fear and extinction memories. Reciprocal intracortical inhibition of these two IC subpopulations gates the emergence of either fear or extinction memory. Using rabies-virus-assisted tracing, we found IC-NAc projection neurons to be preferentially innervated by intercortical inputs from the orbitofrontal cortex (OFC), specifically enhancing extinction to override fear memory. These results demonstrate that IC serves as an operation node harboring distinct projection neurons that decipher fear or extinction memory under the top-down executive control from OFC.

Memories are acquired, stored, and modulated by a distributed neuronal network throughout the brain to guide diverse behaviors. The functional organization of memory-related neuronal ensembles, sets of neurons activated within a short time window related to a particular memory, across different brain areas[1] remains poorly understood. Threat memory is strongly associated with cues of imminent danger and is essential for survival[2]. However, the formation of a new extinction memory through repeated exposure of predictive cues without the threat itself reduces the responses to the conditioned threat,

allowing the animal to adapt flexibly to ever-changing environments[3,4]. The appropriate responses to an identical cue predicting threat or safety, through expression of fear or extinction memory, respectively, are equally important. Fear-related disorders, such as anxiety disorder and posttraumatic stress disorder, are typically characterized by impairments in the extinction memory[2]. Therefore, elucidating the neural circuits engaged in parallel threat versus extinction memories may provide valuable insights for treating these disorders. The neuronal ensembles encoding fear and extinction memories (fear- and

[1]Center for Brain Science, Shanghai Children's Medical Center, Shanghai Jiao Tong University School of Medicine, Shanghai, China. [2]Department of Anatomy and Physiology, Shanghai Jiao Tong University School of Medicine, Shanghai, China. [3]Department of Rehabilitation Medicine, Huashan Hospital, Institute for Translational Brain Research, State Key Laboratory of Medical Neurobiology and Ministry of Education Frontiers Center for Brain Science, Fudan University, Shanghai, China. [4]Department of Integrative Biology and Pharmacology, McGovern Medical School, University of Texas Health Science Center at Houston, Houston, TX, USA. [5]Program in Neuroscience and Mental Health, SickKids Research Institute, Toronto, ON, Canada. [6]Department of Physiology, University of Toronto, Toronto, ON, Canada. [7]Shanghai Research Center for Brain Science and Brain-Inspired Intelligence, Shanghai, China. [8]These authors contributed equally: Qi Wang, Jia-Jie Zhu. ✉e-mail: zhang_siyu@sjtu.edu.cn; liwg@fudan.edu.cn; xu-happiness@shsmu.edu.cn

extinction-memory ensembles) are thought to be located within distinct but intermingled neural circuits, enabling expression of either conditioned fear or extinction responses[3–9]. However, the distribution of these memory ensembles and how the dominant form of memory trace suppresses the other at the network level remain undefined.

The insular cortex (IC) is critical for processing both aversive and appetitive stimuli and coordinating appropriate behavioral responses[10,11], presenting itself as a potential operation node for fear and extinction memories. IC contains anterior and posterior sections[12]. While the posterior IC plays a role in somatosensory, vestibular, and motor integration, the anterior IC has been implicated as the center for interoception[13], which integrates autonomic and visceral information to execute emotional, cognitive, and motivational functions[11]. Furthermore, the functions of anterior IC are reminiscent of those mediated by the amygdala, another well-known harbor for fear and extinction memories. It has been proposed that the amygdala acts like an impulsive system in automatic responses. In contrast, the anterior IC acts as a reflective system in subjective experiences (i.e., subjective feelings)[11,14]. Therefore, the anterior IC stands at a hub to regulate the introduction of subjective feelings into cognitive and motivational processes. However, the roles of IC in fear and extinction memories remain debated[15–18]. Notably, a recent study has shown that IC serves as a crucial state-dependent regulator of fear, setting equilibrium between extinction and fear memories[19]. Nevertheless, whether and how specific populations of IC projection neurons engage in fear and extinction memories remain largely unknown.

Here, using auditory fear conditioning and extinction paradigms to produce both fear and extinction memories in mice, we identified two distinct subpopulations of projection neurons in IC, which target the central amygdala (CeA)[20–23] and nucleus accumbens (NAc)[24–26] to encode fear and extinction memories, respectively. The IC-CeA and IC-NAc projection neurons (IC-CeA and IC-NAc projectors) drive the memory-guided behaviors in opposite directions by reciprocally inhibiting each other. We delineated the whole-brain input maps of IC-CeA and IC-NAc projectors with rabies-virus-assisted tracing. We discovered that among the cortical areas providing different inputs to these IC projection neurons, the orbitofrontal cortex (OFC) specifically enhances extinction memory by activating IC-NAc projectors. Our findings provide compelling evidence exemplifying the segregation of adjacent cortical projection neurons into distinct memory ensembles, which recruit independent subcortical pathways to produce opposite behavioral outcomes associated with the same cue, following the top-down control from specific intercortical connectivity.

## Results

### Identification of fear- and extinction-memory ensembles in anterior IC

To identify neuronal ensembles encoding fear and extinction memories in IC, we employed the method of Fos-targeted recombination in active populations (FosTRAP)[27,28]. Cre-inducible adeno-associated virus (AAV) expressing red fluorescent protein (AAV-EF1α-DIO-mCherry) which expresses CreERT2 under the control of the Fos promoter in a neuronal activity-dependent manner, was injected into IC of FosCreERT2 (FosTRAP2) mice (Fig. 1a). When 4-hydroxytamoxifen (4-OHT) is provided in conjunction with a stimulus, Cre-mediated recombination is enabled transiently and specifically in neurons that were activated during the stimulus period, inducing permanent expression of mCherry in activated neurons within a 6-h window around the 4-OHT injection (Fig. 1b).

To examine fear-memory-dependent activation and reactivation of IC neurons, we subjected these mice to an auditory fear-conditioning and retrieval protocol (Fig. 1c). Fear-conditioning protocol is composed of five variably spaced conditioned tones (conditioned stimulus, CS) and foot shocks (unconditioned stimulus, US) (fear memory, CS + US). The 4-OHT was injected into each mouse

immediately after auditory fear conditioning to capture the neurons activated during fear learning (mCherry+). For fear-memory retrieval, fear-conditioned mice were exposed to the CS (without the US) 3 days after the fear conditioning, and c-fos immunostaining was used to capture the IC neurons activated during fear-memory retrieval (c-fos+, green, Fig. 1d). The control mice went through similar conditioning and retrieval protocols but without foot shocks (CS only). Fear-conditioned mice displayed significantly stronger freezing than control mice to the CS stimuli during fear-memory retrieval (Fig. 1e and Supplementary Fig. 1a). Moreover, we found that most FosTRAPed IC neurons during fear learning were reactivated in fear-memory retrieval (mCherry+ and c-fos+), suggesting that these neurons encoded fear memory (Fig. 1f). These fear-memory ensembles were enriched in the anterior IC (Fig. 1g and Supplementary Fig. 1b). In contrast, the CS-only conditioning and retrieval protocol merely activated few nonoverlapping IC neurons (either mCherry+ or c-fos+) in control mice. These results demonstrated the specificity and efficiency of the FosTRAP method in capturing fear-memory-encoding IC neurons in our experimental paradigm.

Next, we attempted to identify extinction memory ensembles in IC. FosTRAP2 mice with AAV-EF1α-DIO-mCherry injected in IC were subjected to fear conditioning and extinction (Ext.) training (for 2 days after conditioning) before extinction memory retrieval. To capture the activated IC neurons during extinction learning, 4-OHT was injected immediately after the 2nd extinction training session (Fig. 2a, b). These mice were tested for extinction-memory retrieval 3 days after the extinction, and c-fos immunostaining was used to capture the IC neurons activated during the extinction-memory retrieval. A set of control mice were subjected to similar training protocols but without foot shocks (Ext. CS only). Extinction learning significantly reduced the freezing level in the Ext. memory group, although it did not reduce to the same level as that in the Ext. CS only group (Fig. 2c and Supplementary Fig. 1c). In Ext. memory group, the FosTRAPed neurons (activated during extinction learning) highly overlapped with c-fos+ neurons (activated during extinction-memory retrieval). The Ext. CS only conditioning and retrieval protocol merely activated few nonoverlapping IC neurons in control mice. In addition, in another set of control mice, we FosTRAPed the activated neurons during fear retrieval (1 d after fear conditioning) and used c-fos immunostaining to capture the neurons activated during the extinction-memory retrieval. Although fear-retrieval activated IC neurons (FosTRAPed) showed similar distribution patterns as FosTRAPed extinction-memory activated neurons, they rarely overlapped with c-fos+ extinction-memory ensembles (Fig. 2d). Together, these results indicate that FosTRAPed neurons in the Ext. memory group specifically encode extinction memory (Fig. 2d). Notably, the distribution of extinction-memory-encoding neurons was similar to fear-memory-encoding neurons; both were enriched in the anterior IC (Fig. 2e and Supplementary Fig. 1d). These results indicated that the anterior IC is a hub for fear- and extinction-memory ensembles.

We examined the function of IC fear-memory ensembles and extinction-memory ensembles. We first analyzed the potential correlation between the number of FosTRAPed cells and freezing levels during fear retrieval (Fig. 1e) and extinction retrieval (Fig. 2c) tests. As a result, there was no a clear association between freezing levels and the number of FosTRAPed cells during fear or extinction memories (Supplementary Fig. 2), implicating that the size of the IC neuronal ensemble is not the only factor in deciphering the expression of fear or extinction. We then evaluated whether reactivation of these ensembles is necessary for retrieval of fear and extinction memory by inhibiting these IC ensembles during memory retrieval. We injected a Cre-inducible AAV expressing halorhodopsin (NpHR)[29] (AAV-DIO-NpHR-EGFP) into the IC of FosTRAP2 mice to inhibit memory ensembles via illuminating the cell bodies (Supplementary Fig. 3). In control group, a Cre-inducible AAV expressing EGFP (AAV-DIO-EGFP) was injected into

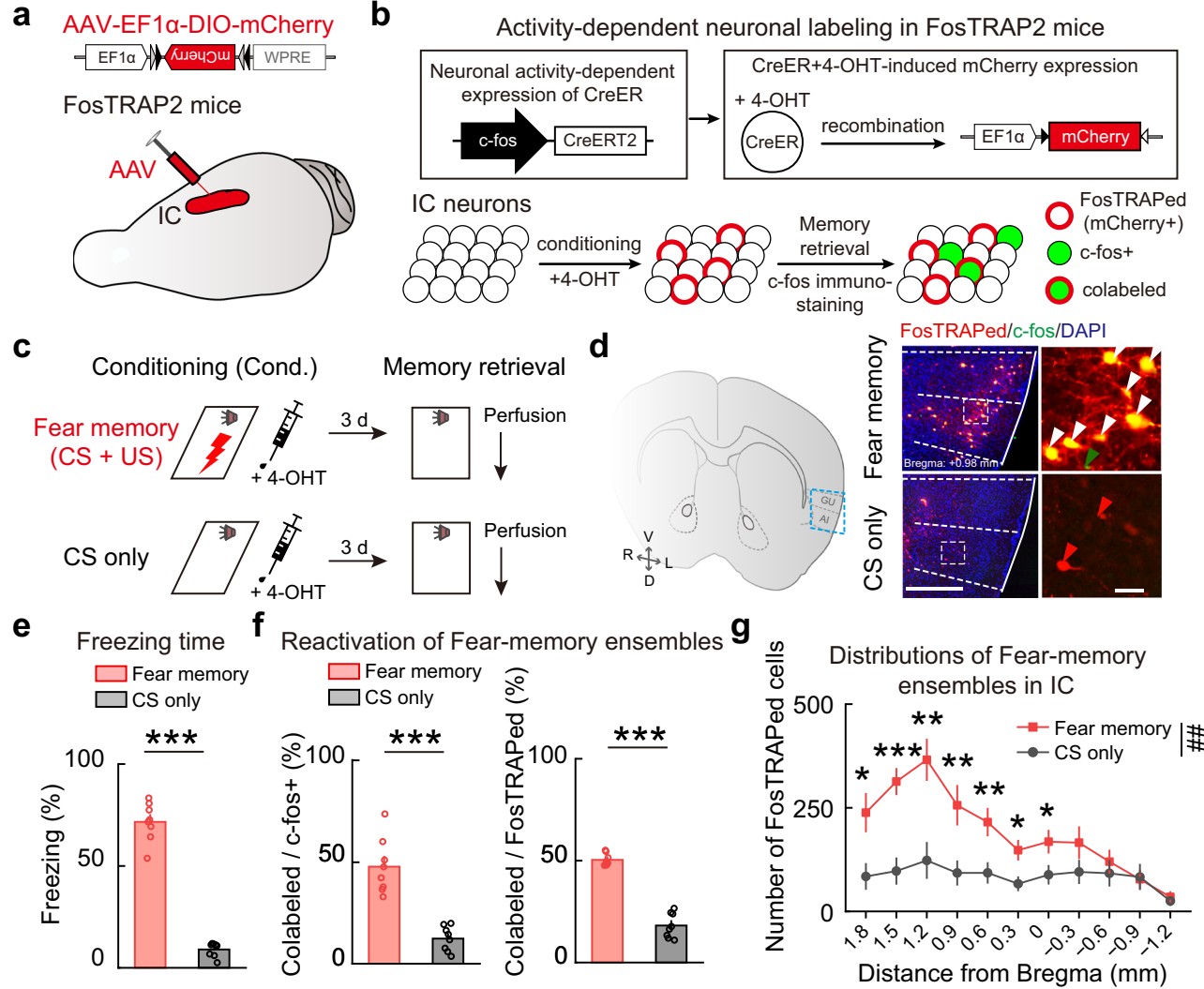

**Fig. 1 | FosTRAPed fear-memory ensembles in IC. a** Schematic of AAV injections to identify memory ensembles in IC. **b** Experimental design. Red circles, FosTRAPed neurons activated during behavioral tagging. Green circles, immunostaining c-fos+ neurons activated during memory retrieval. Yellow circles, colabeled FosTRAPed and c-fos+ neurons. **c**–**g** Distribution patterns of activated IC neurons during fear learning and memory retrieval. Fear memory, $n = 8$ mice; CS only, $n = 8$ mice. **c** Schematic of behavioral protocols. **d** Example images of FosTRAPed (red) and c-fos+ (green) neurons in IC. Left: coronal diagram. Middle: enlarged view of the region in the blue dashed box showing IC. Right: enlarged view of the region in the white-dashed box in IC. Red arrowhead, FosTRAPed only. Green arrowhead, c-fos+ only. White arrowhead, both FosTRAPed and c-fos+. Red, FosTRAPed; green, c-fos+; blue, DAPI. Scale bars, (*left*) 500 μm, (*right*) 50 μm. **e** Freezing responses to the CS during the fear-memory retrieval session ($t_{(14)} = 17.49$, ***$P = 6.5867E{-}11$, two-tailed unpaired Student's *t* test). **f** Overlap rate of IC neurons activated during fear learning and memory retrieval. Left: the percentage of colabeled neurons versus all c-fos+ neurons. $t_{(14)} = 6.676$, ***$P = 1.0502E{-}05$, two-tailed unpaired Student's *t* test. Right: the percentage of colabeled neurons versus all FosTRAPed neurons. $t_{(14)} = 13.44$, ***$P = 2.1653E{-}09$, two-tailed unpaired Student's *t* test. **g** Distribution of FosTRAPed neurons along the anterior-posterior axis of IC. Bregma $+1.80$: $t_{(14)} = 2.836$, *$P = 0.0132$; $+1.50$: $t_{(14)} = 5.000$, ***$P = 1.9452E{-}04$; $+1.20$: $t_{(14)} = 3.770$, **$P = 0.0021$; $+0.90$: $t_{(14)} = 3.005$, **$P = 0.0095$; $+0.60$: $t_{(14)} = 3.073$, **$P = 0.0083$; $+0.30$: $t_{(14)} = 2.931$, *$P = 0.011$; $+0.00$: $t_{(14)} = 2.347$, *$P = 0.0342$; $-0.30$: $t_{(14)} = 1.532$, $P = 0.1478$; $-0.60$: $t_{(14)} = 0.7051$, $P = 0.4923$; $-0.90$: $t_{(14)} = 0.1590$, $P = 0.8759$; $-1.20$: $t_{(14)} = 0.7288$, $P = 0.4781$; two-tailed unpaired Student's *t* test. All: $F_{(1, 14)} = 9.718$; **$P = 0.0076$, two-way repeated measures ANOVA. Data are presented as mean values ± SEM and the error bar represents SEM. Source data are provided as a Source Data file.

the IC of FosTRAP2 mice. Expression of NpHR or EGFP in IC fear-memory and extinction-memory ensembles was achieved in a similar way as in Figs. 1, 2. We then examined the behavioral effects of bilateral optogenetic inhibition of IC fear-memory and extinction-memory ensembles 6 days after behavioral tagging. We found that inhibiting IC fear-memory ensembles reduced freezing responses during fear-memory retrieval (Supplementary Fig. 3c). By contrast, inhibiting IC extinction-memory ensembles significantly impaired extinction memory retrieval, i.e., higher CS-induced freezing responses (Supplementary Fig. 3d). These results demonstrate that IC fear-memory and extinction-memory ensembles play important roles in fear and extinction memories, respectively.

## Fear and extinction memories are segregated in distinct IC projection neurons

To define the spatial distribution of IC neurons associated with the fear and extinction memories, we examined the overlap rate between fear-memory and extinction-memory ensembles. FosTRAP2 mice with AAV-EF1α-DIO-mCherry injected in IC were subjected to fear conditioning and extinction training (Fig. 3a, b). The fear-memory ensembles were FosTRAPed by injecting 4-OHT immediately after fear learning, and the extinction ensembles were identified by c-fos immunostaining after the extinction. We found that the FosTRAPed fear-memory ensembles showed poor overlap with the c-fos+ extinction-memory ensembles (Fig. 3c, d), complementing the above results on the FosTRAPed fear-

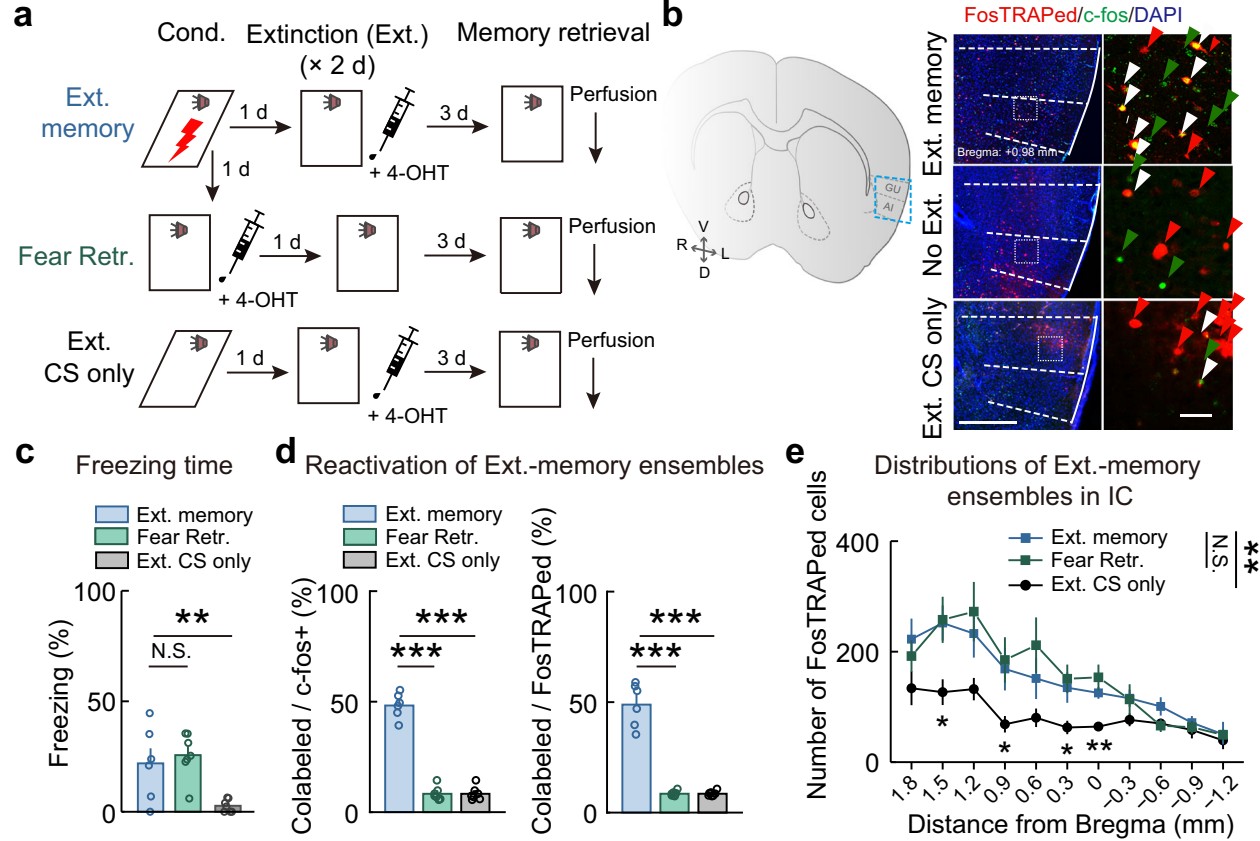

**Fig. 2 | FosTRAPed extinction-memory ensembles in IC.** Similar to Fig. 1, for activated IC neurons during extinction training and memory retrieval. Extinction (Ext.) memory, $n = 6$ mice; fear retrieval without extinction (Fear Retr.), $n = 7$ mice; Ext. CS only, $n = 7$ mice. **a** Schematic of behavioral protocols. **b** Example images of FosTRAPed (red) and c-fos+ (green) neurons in IC. Scale bars, (*left*) 500 μm, (*right*) 50 μm. **c** Freezing responses to the CS. $F_{(2, 17)} = 8.603$, **$P = 0.0026$, one-way ANOVA, main effect of group, followed by Student-Newman-Keuls test. N.S. for Ext. memory vs. Fear Retr., **$P$ for Ext. memory vs. Ext. CS only. **d** Overlap rate of IC neurons activated during extinction training or fear-memory retrieval versus extinction memory retrieval. Left, $F_{(2, 17)} = 176.3$, ***$P = 4.2874E{-}12$, one-way ANOVA, followed by Student-Newman-Keuls test. ***$P$ for Ext. memory vs. Fear Retr., ***$P$ for Ext. memory vs. Ext. CS only. Right, $F_{(2, 17)} = 95.35$, ***$P = 5.7644E{-}10$, one-way

ANOVA, followed by Student-Newman-Keuls test. ***$P$ for Ext. memory vs. Fear Retr., ***$P$ for Ext. memory vs. Ext. CS only. **e** Distribution of FosTRAPed neurons. All: $F_{(2, 17)} = 6.933$, **$P = 0.0063$, two-way repeated measures ANOVA, followed by Bonferroni's test, ***$P < 0.001$ for Ext. memory vs. Ext. CS only, ***$P < 0.001$ for Fear Retr. vs. Ext. CS only, $P = 0.8787$ for Ext. memory vs. Fear Retr. Bregma +1.80: $F_{(2, 17)} = 1.969$, $P = 0.1702$; +1.50: $F_{(2, 17)} = 5.024$, *$P = 0.0193$; +1.20: $F_{(2, 17)} = 3.174$, $P = 0.0674$; +0.90: $F_{(2, 17)} = 3.695$, *$P = 0.0465$; +0.60: $F_{(2, 17)} = 3.210$, $P = 0.0657$; +0.30: $F_{(2, 17)} = 4.582$, *$P = 0.0256$; +0.00: $F_{(2, 17)} = 8.789$, **$P = 0.0024$; −0.30: $F_{(2, 17)} = 1.245$, $P = 0.3128$; −0.60: $F_{(2, 17)} = 2.332$, $P = 0.1274$; −0.90: $F_{(2, 17)} = 0.2374$, $P = 0.7912$; −1.20: $F_{(2, 17)} = 0.1385$, $P = 0.8716$, one-way ANOVA. Data are presented as mean values ± SEM and the error bar represents SEM. Source data are provided as a Source Data file.

retrieval-activated ensembles (Fig. 2d). These results indicate that fear and extinction memory ensembles are segregated in physical proximity along the intracortical axis of IC.

To identify the downstream targets of IC fear- and extinction-memory ensembles, we further examined the axon distribution patterns of FosTRAPed fear-memory and extinction-memory ensembles with Cre-dependent expression of Synaptophysin-tdTomato[30] to label presynaptic boutons (Fig. 3e, f). Interestingly, IC fear-memory ensembles extensively innervated CeA (Fig. 3g, h and Supplementary Fig. 4), which is involved in processing the fear memory[20–23]. In contrast, IC extinction-memory ensembles preferentially innervated NAc (Fig. 3g, h, i and Supplementary Fig. 4), which is related to extinction memory[24–26]. These results suggested that distinct IC projection neurons respectively process fear and extinction memories, likely through activating different downstream targets in subcortical regions.

### Nonoverlapping IC-CeA and IC-NAc projectors encode fear and extinction memories

Since the IC fear- and extinction-memory ensembles preferentially innervate CeA and NAc, we hypothesized that IC-CeA and IC-NAc projectors encode fear and extinction memories, respectively. Retrograde tracing showed that IC-CeA and IC-NAc projectors represent

distinct neuronal populations (Supplementary Fig. 5a–d), consistent with a previous study[17]. Furthermore, the IC-CeA and IC-NAc projectors are distributed across different cortical layers. Most IC projectors were found in layer 5, with more IC-CeA projectors in layer 2/3 and more IC-NAc projectors in layer 6 (Supplementary Fig. 5e).

To investigate the function of IC-CeA and IC-NAc projectors, we monitored behavior-related neuronal activities of these neurons in fear and extinction memories via fiber photometric measurements of calcium transients. To express the calcium indicator into the IC-CeA and IC-NAc projectors, Cre-inducible AAV expressing GCaMP6m (AAV-EF1α-DIO-GCaMP6m) was injected in IC, and the retrograde AAV expressing Cre (Retro-AAV-Syn-Cre-mCherry) was injected into CeA or NAc of wild-type mice (Fig. 4a and Supplementary Fig. 6). These mice were subjected to a training protocol including fear conditioning, fear-memory retrieval, extinction, and extinction-memory retrieval, showing similar behavioral responses between the IC-CeA and IC-NAc groups (Fig. 4b, c). The freezing responses to the CS gradually increased during the fear conditioning phase, kept at a high level during the fear retrieval, and then decreased during the extinction and extinction-memory retrieval. Fluorescence changes in GCaMP6m calculated by either ΔF/F (Fig. 4d–g) or z-score (Supplementary Fig. 7) indicated strong behavior-related activities specific to

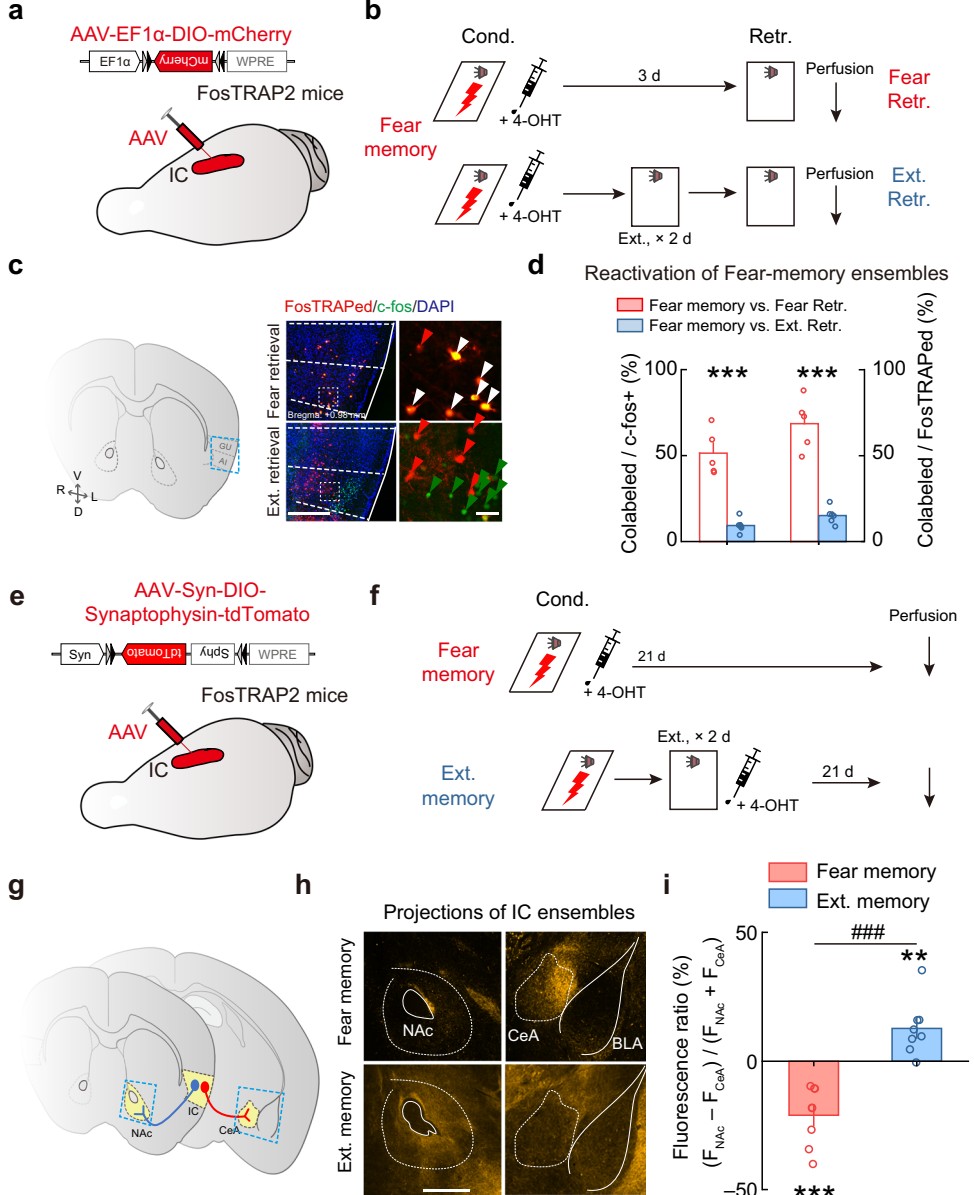

**Fig. 3 | Fear and extinction memories activate distinct IC projection neurons.**
**a–d** Comparison between the distribution patterns of activated IC neurons during
fear learning versus fear or extinction memory retrieval. **a** Schematic of AAV
injection in IC. **b** Schematic of behavioral protocols. **c** Example images of Fos-
TRAPed (red) and c-fos+(green) neurons in IC. Left: coronal diagram. Middle:
enlarged view of the region in the blue dashed box showing IC. Right: enlarged view
of the region in the white-dashed box in IC. Red arrowhead, FosTRAPed only. Green
arrowhead, c-fos+ only. White arrowhead, both FosTRAPed and c-fos+. Red, Fos-
TRAPed; green, c-fos+; blue, DAPI. Scale bars, (*left*) 500 μm, (*right*) 50 μm.
**d** Overlap rate of IC neurons activated during fear learning versus fear or extinction
memory retrieval. The overlap rate in the fear learning versus fear-memory retrieval
group (red bar) was significantly higher than that in the fear learning versus
extinction memory retrieval group (blue bar). Fear retrieval, $n = 5$ mice. Ext.
retrieval, $n = 6$ mice. Colabeled/c-fos+: $t_{(9)} = 7.508$, ***$P = 3.6629E-05$; Colabeled/

FosTRAPed: $t_{(9)} = 8.253$, ***$P = 1.7238 E-05$, two-tailed unpaired Student's $t$ test.
**e–i** Distinct downstream targets of IC fear- and extinction-memory ensembles.
**e** Schematic of AAV injections in IC. **f** Schematic of behavioral protocols.
**g** Schematic diagram of IC fear- and extinction-memory ensembles with distinct
projections. NAc nucleus accumbens, CeA central amygdala. **h** Example images of
the axons from the IC fear- (*upper*) and extinction-(*lower*) memory-associated
neurons in NAc (*left*) and CeA (right). BLA basolateral amygdala. Scale bars, 200 μm.
**i** Comparison of fluorescence ratios of the axons from IC fear- and extinction-
memory ensembles in NAc and CeA. Each group, $n = 8$ mice. Fear memory vs.
extinction memory: $t_{(14)} = 6.083$, ###$P = 2.8253E-05$; fear memory vs. the value of
zero, $t_{(14)} = 5.192$, ***$P = 1.37E-04$; extinction memory vs. the value of zero,
$t_{(14)} = 3.349$, **$P = 0.0048$, two-tailed unpaired Student's $t$ test. Data are presented as
mean values ± SEM and the error bar represents SEM. Source data are provided as a
Source Data file.

IC-CeA and IC-NAc projectors. Although both IC-CeA and IC-NAc
projectors showed strong responses to foot shocks (US), their
responses to the auditory cues (CS) differed. During fear condition-
ing (CS-US pairing), CS markedly increased the Ca²⁺ response in IC-
CeA projectors, with little change in IC-NAc projectors (Fig. 4d, f and
Supplementary Fig. 8a, c). Furthermore, the IC-CeA projectors were
preferentially activated by fear-memory retrieval, whereas the IC-

NAc projectors selectively responded to extinction-memory retrieval
(Fig. 4e, g and Supplementary Fig. 8b, d). As a control, we did not
detect behavior-related changes in fluorescent signals in IC-CeA or
IC-NAc projectors of mice that expressed mCherry (Supplementary
Fig. 9). Together, these results indicated that nonoverlapping IC-CeA
and IC-NAc projectors encode fear and extinction memories,
respectively.

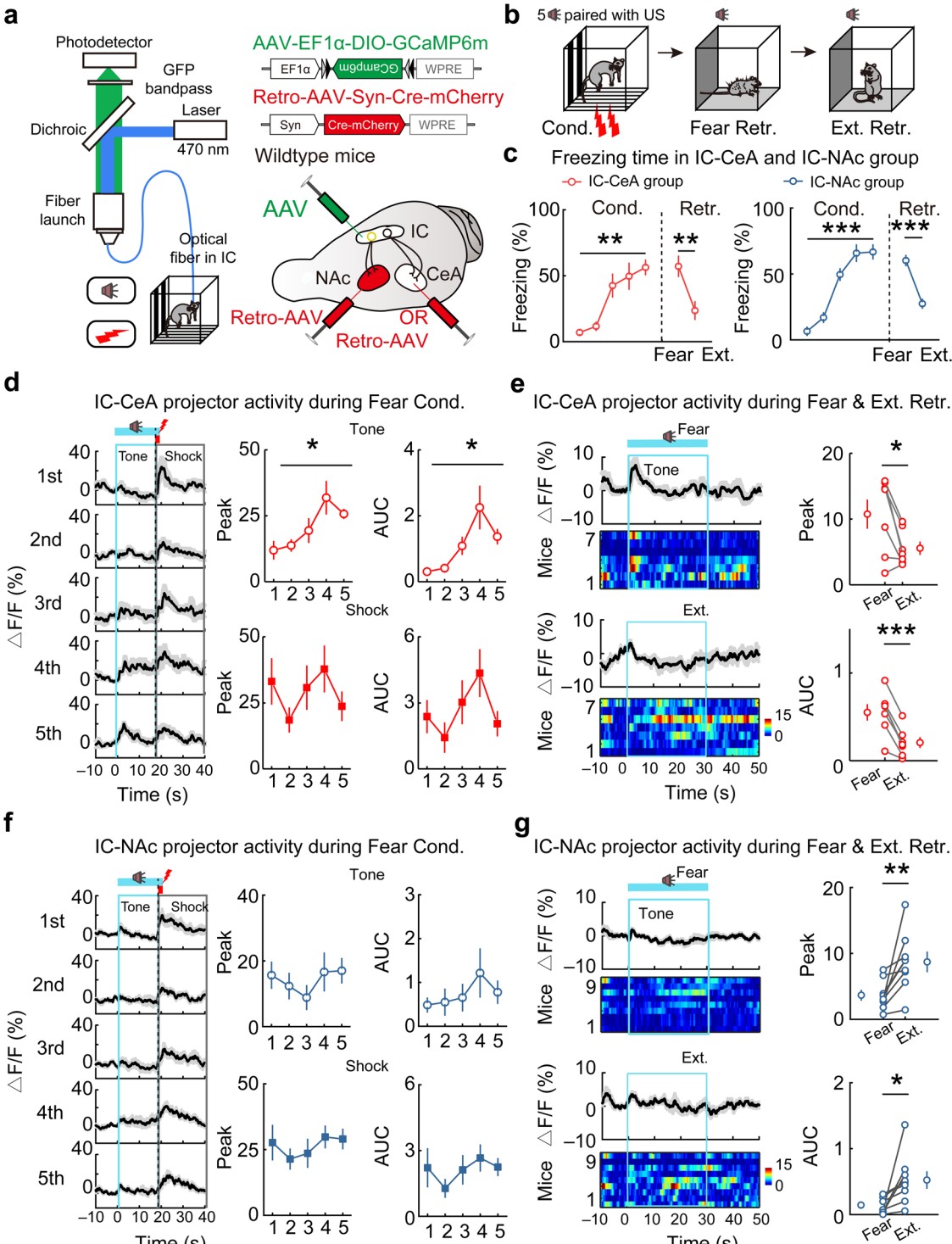

**Fig. 4 | Distinct response patterns of IC-CeA and IC-NAc projectors to fear learning and fear- or extinction-memory retrieval. a** Schematic diagram. **b** Behavioral protocols. **c** Freezing responses. IC-CeA projector group, $n = 7$ mice; IC-NAc projector group, $n = 9$ mice. IC-CeA, Cond.: $F_{(1.822, 10.93)} = 14.40$, **$P = 0.0010$, one-way repeated-measure ANOVA; Retr.: $t_{(6)} = 5.338$, **$P = 0.0018$; two-tailed paired Student's $t$ test; IC-NAc, Cond.: $F_{(2.490, 19.92)} = 51.50$, ***$P < 0.0001$, one-way repeated-measure ANOVA; Retr.: $t_{(8)} = 6.674$, ***$P = 0.0002$; two-tailed paired Student's $t$ test. **d**–**g** Calcium signals (ΔF/F). **d**, **f** Left: Average calcium signals aligned to the onset of the CS during fear conditioning. Thick lines, mean; shaded areas, SEM. Right: the peak and the area under the curves (AUC) during tone (in cyan box) and shock (in gray box). **d** Tone peak: $F_{(1.932, 11.59)} = 4.688$, *$P = 0.0331$; tone AUC: $F_{(1.358, 8.147)} = 6.409$, *$P = 0.0283$; shock peak: $F_{(1.303, 7.818)} = 1.508$, $P = 0.2663$; shock AUC: $F_{(1.803, 10.82)} = 2.373$, $P = 0.1426$, one-way repeated-measure ANOVA. **f** Tone peak:

$F_{(1.955, 15.64)} = 0.6798$, $P = 0.5179$; tone AUC: $F_{(1.956, 15.65)} = 0.7962$, $P = 0.4659$; shock peak: $F_{(2.218, 17.75)} = 0.8455$, $P = 0.7730$; shock AUC: $F_{(1.813, 14.51)} = 0.8906$, $P = 0.4222$, one-way repeated-measure ANOVA. **e**, **g** Average calcium signals aligned to the onset of the CS during fear- or extinction-memory retrieval. Left upper: calcium signals during fear memory retrieval, average calcium signals and heatmap of calcium signals in each mouse. Left lower: similar as above for the calcium signals during extinction memory retrieval. Right: The peak and AUC during tone in fear- and extinction-memory retrieval (in cyan box). **e** Peak: $t_{(6)} = 3.105$, *$P = 0.021$; AUC: $t_{(6)} = 7.325$, ***$P = 3.3071E-04$, two-tailed paired Student's $t$ test. **g** Peak: $t_{(8)} = 3.450$, **$P = 0.0087$; AUC: $t_{(8)} = 2.839$, *$P = 0.0218$; two-tailed paired Student's $t$ test. Data are presented as mean values ± SEM and the error bar represents SEM. Source data are provided as a Source Data file.

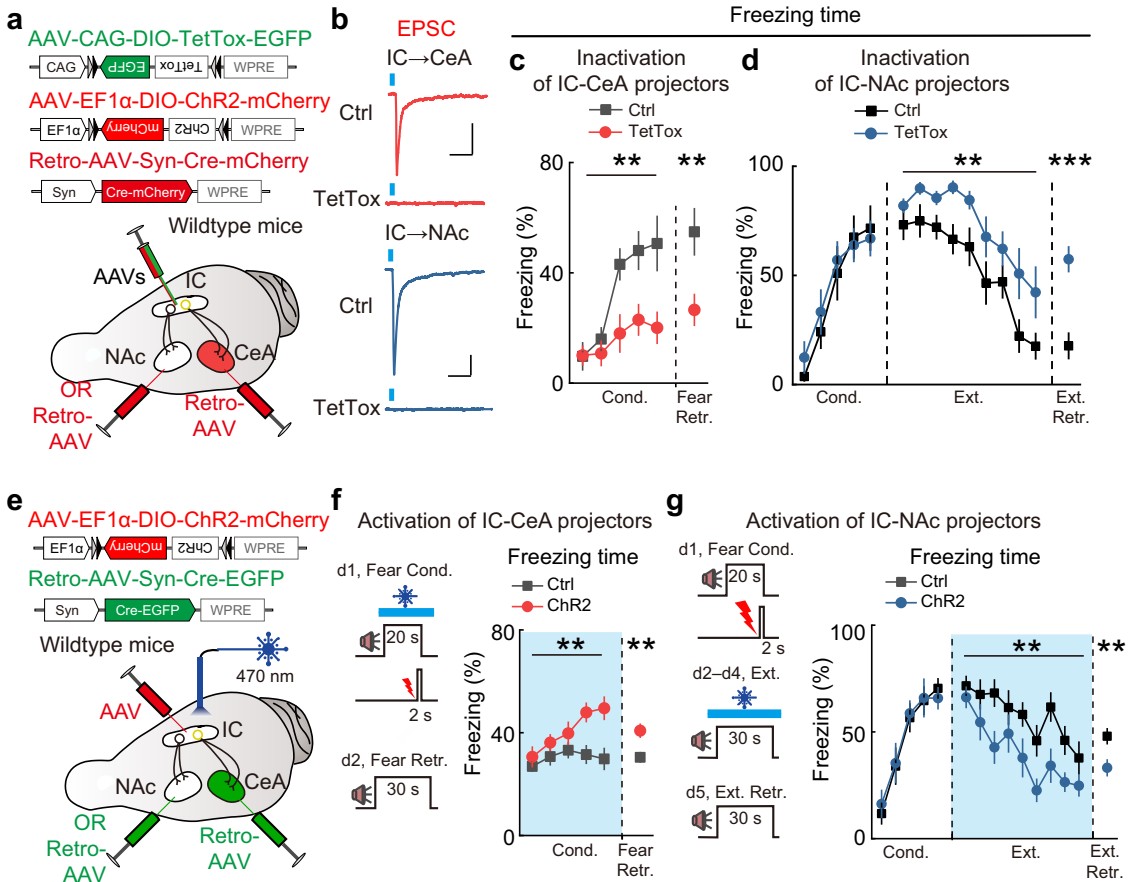

**Fig. 5 | Effects of genetic silencing and optogenetic activation of IC-CeA or IC-NAc projectors on fear and extinction memories. a–d** Effects of genetic silencing of IC-CeA or IC-NAc projectors. **a** Schematic of AAV injections. **b** Patch-clamp recordings from the neurons in CeA and NAc. Scale bars, 100 ms, 50 pA. **c** Effects of genetic silencing of IC-CeA projectors. Ctrl (EGFP), $n = 15$ mice, TetTox, $n = 16$ mice. Cond.: $F_{(1, 29)} = 8.733$, **$P = 0.0061$, two-way repeated measures ANOVA; Retr.: $t_{(29)} = 2.853$, **$P = 0.0079$, two-tailed unpaired Student's $t$ test. **d** Effects of genetic silencing of IC-NAc projectors. Ctrl (EGFP), $n = 10$ mice, TetTox, $n = 10$ mice. Cond.: $F_{(1, 18)} = 0.1298$, $P = 0.7228$, two-way repeated measures ANOVA; Ext.: $F_{(1, 18)} = 8.519$, **$P = 0.0092$, two-way repeated measures ANOVA; Retr.: $t_{(18)} = 4.944$, ***$P = 1.0482E$ $−04$, two-tailed unpaired Student's $t$ test. **e–g** Effects of optogenetic activation of IC-CeA or IC-NAc projectors. **e** Schematic of AAV injections. **f** Effects of optogenetic

activation of IC-CeA projectors. Left: optogenetic activation and behavioral protocols. Right: freezing responses to the CS during fear conditioning and fear-memory retrieval. Blue box, light-on period. Ctrl (mCherry), $n = 9$ mice, ChR2, $n = 9$ mice. Cond.: $F_{(1, 16)} = 14.57$, **$P = 0.0015$, two-way repeated measures ANOVA; Retr.: $t_{(16)} = 3.816$, **$P = 0.0079$, two-tailed unpaired Student's $t$ test. **g** Similar to (**f**) for the effects of optogenetic activation of IC-NAc projectors. Ctrl (mCherry), $n = 11$ mice, ChR2, $n = 11$ mice. Cond.: $F_{(1, 20)} = 0.01930$, $P = 0.8909$, two-way repeated measures ANOVA; Ext.: $F_{(1, 20)} = 11.93$, **$P = 0.0025$, two-way repeated measures ANOVA; Retr.: $t_{(20)} = 2.924$, **$P = 0.0084$, two-tailed unpaired Student's $t$ test. Data are presented as mean values ± SEM and the error bar represents SEM. Source data are provided as a Source Data file.

## Subcortical pathways from IC-CeA and IC-NAc projectors differentially regulate fear and extinction memories

To directly test the causal relationship between the activity of IC projection neurons and memory maintenance, we bidirectionally modulated synaptic release of IC-CeA and IC-NAc projectors. To silence IC-CeA and IC-NAc projectors, Cre-inducible AAV expressing the light chain of tetanus toxin (TetTox, AAV-DIO-TetTox-EGFP)[31] was injected in IC, and the retrograde AAV expressing Cre (Retro-AAV-Syn-Cre-mCherry) was injected into CeA or NAc of wild-type mice (Fig. 5a). To confirm the inactivation efficiency, Cre-inducible AAV expressing channelrhodopsin-2 (ChR2, AAV-DIO-ChR2-mCherry) was co-injected into the IC. The expression of TetTox in IC-CeA and IC-NAc projectors significantly reduced the ChR2-mediated excitatory postsynaptic currents (oEPSCs) in CeA and NAc neurons evoked by optogenetic activation of IC axon terminals, compared with the control mice injected with AAV-DIO-EGFP (expressing EGFP without TetTox) (Fig. 5b). We then examined the behavioral effects of silencing IC-CeA and IC-NAc projectors. We found that silencing IC-CeA projectors markedly reduced freezing responses during fear conditioning and fear-memory retrieval (Fig. 5c). By contrast, silencing IC-NAc projectors did not

affect fear conditioning but significantly impaired fear extinction, showing a slower decrease in CS-induced freezing responses (Fig. 5d). As a control for locomotor activity, silencing IC-CeA or IC-NAc projectors did not affect the behaviors in open field test (Supplementary Fig. 10).

Moreover, optogenetic inactivation of IC-CeA and IC-NAc projectors induced similar behavioral effects as TetTox. While optogenetic inactivation of IC-CeA projectors during fear conditioning significantly reduced freezing responses during fear conditioning and fear memory retrieval, optogenetic inactivation of IC-NAc projectors during extinction markedly impaired fear extinction, showing a slower decline in CS-induced freezing responses (Supplementary Fig. 11). To further clarify the effects of IC-CeA and IC-NAc projectors on memory expression, we limited our optogenetic manipulations to the expression test period and found significant inhibitory effects on fear and extinction memory retrieval, respectively (Supplementary Fig. 12). Again, optogenetic inactivation of IC-CeA or IC-NAc projectors did not affect the behaviors in open field test (Supplementary Fig. 13). These results further support the notion that IC-CeA and IC-NAc projectors play important roles in fear conditioning and extinction, respectively.

To further test the functional significance of IC-CeA and IC-NAc projectors, we applied optogenetic activation of these projection neurons (Fig. 5e and Supplementary Fig. 14). Without fear conditioning, activation of IC-CeA projectors did not induce any detectable freezing behavior (Supplementary Fig. 15a). Activation of IC-CeA projectors during fear conditioning with moderate (0.3 mA) but not strong (0.5 mA) foot shocks enhanced the freezing behavior, and this enhancement was maintained in the fear-memory retrieval (Fig. 5f and Supplementary Fig. 15b). These results suggest that activation of IC-CeA projectors facilitates the formation and expression of fear memory. By contrast, activation of IC-NAc projectors during fear conditioning did not affect freezing behavior (Supplementary Fig. 16). Instead, activation of IC-NAc projectors during extinction reduced freezing responses, and this reduction was maintained in the extinction-memory retrieval (Fig. 5g). Optogenetic activation of IC-CeA and IC-NAc projectors did not affect the locomotor activity in open field test (Supplementary Fig. 17). Together, these results demonstrate that IC-CeA and IC-NAc projectors specifically contribute to fear and extinction memories, respectively.

## IC-CeA and IC-NAc projectors differentially contribute to memory-associated valence processing

We next examined whether IC-CeA and IC-NAc projectors contribute to memory-associated valence processing through optogenetic activation of IC projectors during conditioned place aversion and preference (CPA and CPP) experiments using mice that had acquired fear or extinction memory before (Supplementary Fig. 18a, b). In the CPA and CPP assays, one chamber of the apparatus was assigned as memory-associated valence-paired (optogenetic activation of IC projectors in mice without or with prior memory acquisition) and the other as unpaired (without light stimulation). With optogenetic activation of IC-CeA projectors, naïve mice failed to develop either avoidance or preference to the light-paired chamber after 3 days of conditioning (Supplementary Fig. 18c–g). However, fear-memory acquired mice exhibited an apparent avoidance of this chamber, showing less time spent during the post-training than pre-training test, with a decrease of the normalized CPA score (Supplementary Fig. 18h–l). Conversely, with optogenetic activation of IC-NAc projections, which conditioning in naïve mice did not (Supplementary Fig. 18m–q), that in extinction-memory acquired mice (Supplementary Fig. 18r–v) established a CPP for the light-paired chamber, showing more time spent during the post-training than pre-training test, with a tendency to increase of the normalized CPP score. Together, these results supported the notion that IC-CeA and IC-NAc projectors contribute to the fear-memory-associated aversion and extinction-memory-associated preference, respectively.

## Reciprocal inhibition between IC-CeA and IC-NAc projectors via intracortical interneurons

To fully understand how fear and extinction memories specifically recruit distinct projection neurons in IC, we examined the intrinsic electrophysiological properties of IC-CeA and IC-NAc projectors and the potential interactions between them in local circuits with fear conditioning and extinction paradigms. To test the intrinsic properties of IC-CeA and IC-NAc projectors identified by retrograde AAV expressing green and red fluorescent proteins injected in CeA and NAc, we performed whole-cell recording on these projection neurons in IC slices (Fig. 6a). We focused on layer 5 because most IC projectors are located here. IC-CeA projectors have significantly lower rheobase (Fig. 6b) and higher intrinsic excitability (Supplementary Fig. 19) than IC-NAc projectors, indicating that IC-CeA projectors are more excitable than IC-NAc projectors.

To investigate the interaction between IC-CeA and IC-NAc projectors, the retrograde AAVs were used to express ChR2 in either population of projectors and EGFP in the other. For example, to express ChR2 in IC-CeA projectors, the Cre-inducible AAV expressing ChR2 (AAV-DIO-ChR2-mCherry) was injected in IC, and the retrograde AAV expressing Cre (Retro-AAV-Syn-Cre) was injected in CeA. Simultaneously, the retrograde AAV expressing EGFP (Retro-AAV-Syn-EGFP) was injected in NAc to label IC-NAc projectors with green fluorescence (Fig. 6c). Optogenetic activation of IC-CeA projectors induced both excitatory and inhibitory postsynaptic currents (oEPSC and oIPSC) on IC-NAc projectors. The oEPSCs showed short onset latencies ($2.5 \pm 0.3$ ms, mean $\pm$ SEM), suggesting monosynaptic excitatory inputs. However, the oIPSCs showed significantly longer latencies ($5.7 \pm 0.2$ ms, mean $\pm$ SEM) and were completely blocked by CNQX (20 μM) plus D-APV (50 μM), suggesting disynaptic inhibition (Supplementary Fig. 20). We then measured the changes of excitatory and inhibitory inputs with fear conditioning on IC-NAc projectors induced by IC-CeA-projector activation (Fig. 6c, d). After fear conditioning, the paired-pulse ratios (PPRs) of oEPSCs significantly increased, suggesting that fear conditioning decreases the presynaptic release probability at the synapses of IC-CeA projectors onto IC-NAc projectors (Fig. 6d). Next, we examined adaptive changes of IC-CeA projector-mediated disynaptic inhibition on IC-NAc projectors. At a depolarized membrane potential (−40 mV), activation of IC-CeA projectors induced biphasic synaptic responses in IC-NAc projectors, comprised of an early EPSC and a delayed IPSC. Remarkably, compared to the CS-only condition group, fear conditioning resulted in an increased IPSC-to-EPSC ratio in the inputs received by IC-NAc projectors with activation of IC-CeA projectors (Fig. 6d). These results indicate that fear conditioning changes the excitation-inhibition balance in local circuits, i.e., IC-NAc projectors receive less excitatory and more inhibitory inputs with IC-CeA-projector activation.

Using similar approaches, we measured the synaptic inputs on IC-CeA projectors induced by the activation of IC-NAc projectors. Similar to the inputs from IC-CeA to IC-NAc projectors, activation of IC-NAc projectors induced a monosynaptic oEPSC (onset latency, $2.6 \pm 0.2$ ms, mean $\pm$ SEM) and a disynaptic oIPSC ($6.0 \pm 0.2$ ms, mean $\pm$ SEM) on IC-CeA projectors (Supplementary Fig. 20). We then examined the adaptive changes of IC-NAc-projector-mediated excitatory and inhibitory inputs on IC-CeA projectors with extinction (Fig. 6e, f). After extinction, compared with fear retrieval, the PPRs of oEPSCs significantly increased, suggesting that extinction decreases the presynaptic release probability at the synapses of IC-NAc projectors onto IC-CeA projectors (Fig. 6f). Moreover, the IPSC-to-EPSC ratio significantly increased after extinction in the IC-CeA projectors induced by IC-NAc-projector activation (Fig. 6f), consistent with our observation that IC-CeA projectors were suppressed after extinction. Together, these results demonstrate that the IC-CeA and IC-NAc projectors reciprocally inhibit each other via the recruitment of local inhibitory neurons. Especially, the inhibition from IC-CeA to IC-NAc projectors increased after fear conditioning. In contrast, IC-NAc projectors provided more inhibition to IC-CeA projectors after extinction. These results provide the mechanistic basis at the cellular level for active circuit competition between fear and extinction memories in IC.

## Intercortical connectivity between the orbitofrontal cortex (OFC) and IC projectors gates extinction learning and memory

To map the monosynaptic inputs to IC-CeA and IC-NAc projectors, we used the rabies-virus (RV)-mediated retrograde tracing. We injected retrograde AAV expressing Cre (Retro-AAV-Syn-Cre) into CeA or NAc of wild-type mice, and AAVs with Cre-inducible expression of avian-specific retroviral receptor (TVA) and rabies glycoprotein (RG) into IC. Four weeks after these AAV injections, RV expressing DsRed (RV-ENVA-dG-DsRed) was injected into IC (Fig. 7a). This viral strategy ensured that TVA and RG were expressed only in the IC-CeA or IC-NAc projectors retrogradely labeled with Cre recombinase, thus restricting RV labeling to their presynaptic inputs (Supplementary Fig. 21a, b). The starter cells (expressing both EGFP and DsRed) showed a comparable

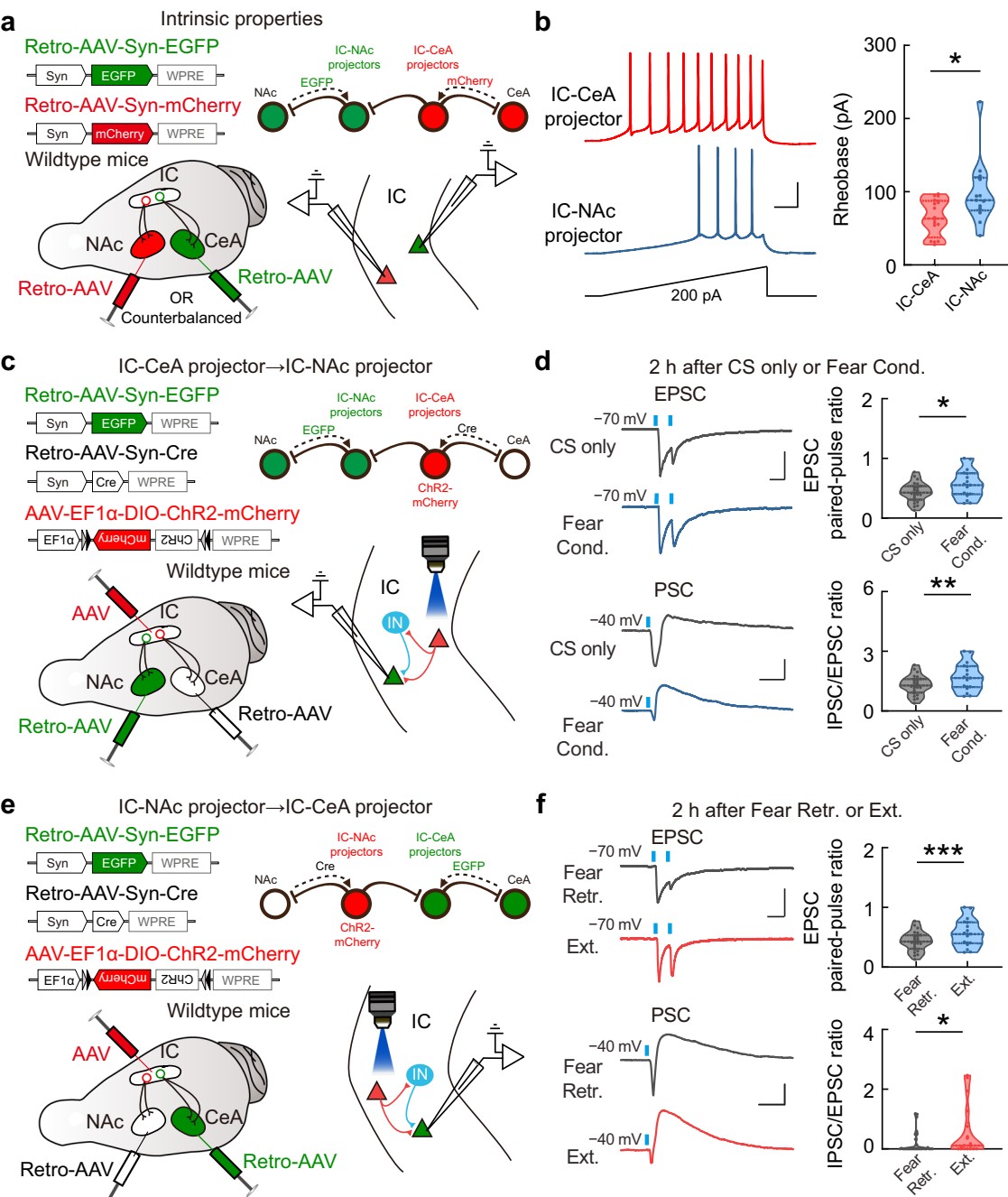

**Fig. 6 | Interactions between IC-CeA and IC-NAc projectors in the local circuits changed with fear and extinction memories. a, b** Intrinsic electrophysiological properties. **a** Schematic of slice recordings. **b** Ramping current injection induced-action potentials. Left: example traces. Scale bars, 50 ms, 200 pA. Right: graphic summary. IC-CeA projector, $n = 16$ neurons from 6 mice; IC-NAc projector, $n = 13$ neurons from 7 mice. $t_{(27)} = 2.572$, *$P = 0.0159$, two-tailed unpaired Student's $t$ test. **c, d** Effects of activation of IC-CeA projectors on IC-NAc projectors. **c** Schematic of AAV injections. Whole-cell recordings were made on IC-NAc projectors while activating IC-CeA projectors via blue-light stimuli. **d** Upper left: example traces of oEPSCs (holding = −70 mV) at the synapses from IC-CeA to IC-NAc projectors. Scale bars, 50 ms, 200 pA. Upper right: statistical analysis of the paired-pulse ratio of oEPSCs. CS only, $n = 24$ neurons from 9 mice; Fear Cond., $n = 18$ neurons from 8 mice. $t_{(40)} = 2.197$, *$P = 0.0339$, two-tailed unpaired Student's $t$ test. Lower left:

example traces showing oPSCs (holding = −40 mV) in IC-NAc projectors driven by activation of IC-CeA projectors. Scale bars, 50 ms, 200 pA. Lower right: statistical analysis of the IPSC/EPSC ratio of oPSCs. CS only, $n = 16$ neurons from 8 mice; Fear Cond., $n = 12$ neurons from 8 mice. $t_{(26)} = 3.454$, **$P = 0.0019$, two-tailed unpaired Student's $t$ test. Blue dots, blue-light stimuli. Scale bars, 50 ms, 200 pA. **e, f** Similar to (**c, d**) for the EPSCs and PSCs in IC-CeA projectors driven by activation of IC-NAc projectors. **f** Upper left: scale bars, 50 ms, 200 pA. Lower left: scale bars, 50 ms, 100 pA. Paired-pulse ratios of EPSC, Fear Retr., $n = 20$ neurons from 10 mice; Ext., $n = 21$ neurons from 9 mice. IPSC/EPSC ratios, Fear Retr., $n = 30$ neurons from 11 mice; Ext., $n = 18$ neurons from 9 mice. Upper: $t_{(39)} = 4.291$, ***$P = 1.1363E{-}04$; Lower: $t_{(46)} = 2.109$, *$P = 0.0404$, two-tailed unpaired Student's $t$ test. Source data are provided as a Source Data file.

laminar distribution to the IC-CeA and IC-NAc projectors previously shown to be labeled by retrograde AAVs only (Supplementary Fig. 5e). Both sets of starter cells were located in layer 5, with abundant starter cells for IC-CeA projectors also found in layer 2/3 and that for IC-NAc

projectors found in layer 6 (Supplementary Fig. 21c). Trans-synaptically-labeled presynaptic neurons (expressing DsRed only) of both IC-CeA and IC-NAc projectors were found in multiple cortical and subcortical regions with a majority of them located within the

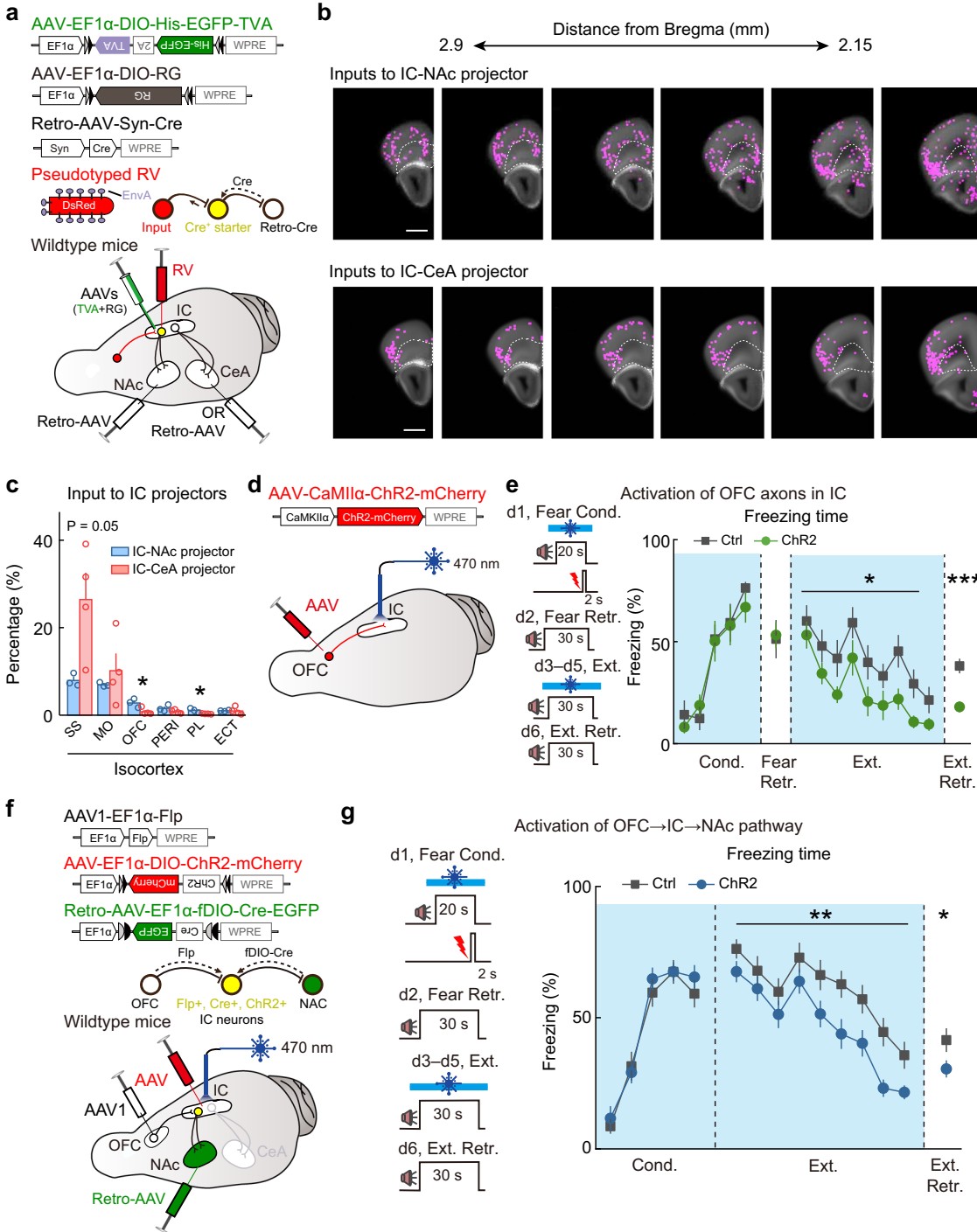

**Fig. 7 | Activation of OFC → IC → NAc pathway promotes extinction learning and memory. a–c** Whole-brain mapping of monosynaptic inputs to IC projectors. **a** Viral injection strategy. **b** Distributions of RV-labeled input neurons. Each dot represents one RV-labeled neuron. White-dashed curves denote outlines of OFC. Scale bar, 1 mm. **b** Was created with Matlab (R2014a, USA, https://www.mathworks.com) by aligning fluorescence imaging to the fully open-access corresponding coronal section of Allen Mouse Brain Atlas and Allen Reference Atlas - Mouse Brain. Allen Mouse Brain Atlas, mouse.brain-map.org and atlas.brain-map.org. **c** Quantification of labeled monosynaptic inputs to IC projectors. IC-NAc projectors, $n = 3$ mice; IC-CeA projectors, $n = 4$ mice. SS somatosensory area, MO somatomotor area, OFC orbitofrontal cortex, PERI perirhinal area, PL prelimbic area, ECT ectorhinal area. SS: $t_{(5)} = 2.538$, $P = 0.052$; MO: $t_{(5)} = 0.6862$, $P = 0.5231$; OFC: $t_{(5)} = 2.954$, *$P = 0.0317$; PERI: $t_{(5)} = 1.646$, $P = 0.1608$; PL: $t_{(5)} = 3.212$, *$P = 0.0237$; ECT: $t_{(5)} = 0.1192$, $P = 0.9098$, two-tailed unpaired Student's $t$ test. **d, e** Effects of optogenetic activation of OFC → IC projections. **d** Schematic of AAV injection and optic fiber implantation. **e** Left: optogenetic activation and behavioral protocols. Right:

freezing responses to the CS. Blue box, light-on period. Ctrl (mCherry), $n = 9$ mice; ChR2, $n = 9$ mice. Cond.: $F_{(1, 16)} = 0.09711$, $P = 0.7593$, two-way repeated measures ANOVA; Fear Retr.: $t_{(16)} = 0.1854$, $P = 0.8552$, two-tailed unpaired Student's $t$ test; Ext.: $F_{(1, 16)} = 5.940$, *$P = 0.0269$, two-way repeated measures ANOVA; Ext. Retr.: $t_{(16)} = 5.129$, ***$P = 1.0098E–04$, two-tailed unpaired Student's $t$ test. **f** Schematic of AAV injection for expressing ChR2 in IC-NAc projectors dedicated to receiving long-range projections from the OFC. Optic fiber was implanted in IC. **g** Effects of optogenetic activation of IC-NAc projectors dedicated to receiving long-range inputs from OFC. Left: optogenetic activation and behavioral protocols. Right: freezing responses to the CS. Blue box, light-on period. Ctrl (mCherry), $n = 18$ mice, ChR2, $n = 20$ mice. Cond.: $F_{(1, 36)} = 0.3609$, $P = 0.5518$, two-way repeated measures ANOVA; Ext.: $F_{(1, 36)} = 8.020$, **$P = 0.0075$, two-way repeated measures ANOVA; Ext. Retr., $t_{(36)} = 2.128$, *$P = 0.0402$, two-tailed unpaired Student's $t$ test. Data are presented as mean values ± SEM and the error bar represents SEM. Source data are provided as a Source Data file.

isocortex (Fig. 7b, c). Within the cortical areas, IC-NAc projectors received more extensive top-down inputs from the frontal cortices (orbitofrontal cortex, OFC, and prelimbic area, PL) than IC-CeA projectors, and the inputs from OFC were about three times more than those from PL. By contrast, IC-CeA projectors received more inputs ($P = 0.05$, two-tailed unpaired Student's $t$ test, IC-CeA versus IC-NAc) from the somatosensory cortex (SS). In the subcortical areas, IC-NAc projectors received more inputs from the anterior part of the basolateral amygdalar nucleus (BLAa), whereas IC-CeA projectors received more inputs from CeA (Supplementary Fig. 22).

We next examined the intercortical connectivity of the top-down inputs from OFC to IC-NAc and IC-CeA projectors using whole-cell recordings in IC slices. We injected AAV-CaMKIIα-ChR2-mCherry into OFC to express ChR2 in OFC projection neurons, and injected Retro-AAV-Syn-EGFP into the NAc or CeA to retrogradely express EGFP in IC-NAc or IC-CeA projectors. The whole-cell recording was performed on IC-NAc and IC-CeA projectors while optogenetically activating OFC axon terminals in IC (Supplementary Fig. 23a, b). We observed both oEPSCs and disynaptic oIPSCs in IC-NAc and IC-CeA projectors (Supplementary Fig. 23c–e). Notably, all of the recorded IC-NAc projectors (100%, 18/18) received direct excitatory inputs from OFC, whereas only 69% (11/16) of IC-CeA projectors showed oEPSCs in response to activation of OFC axon terminals, supporting more abundant OFC inputs to IC-NAc than IC-CeA projectors (Supplementary Fig. 23d). Furthermore, the IPSC-to-EPSC ratio of OFC-mediated inputs on IC-NAc projectors was lower than that on IC-CeA projectors (Supplementary Fig. 23f). These results indicate that OFC provides more excitation to IC-NAc projectors and more inhibition to IC-CeA projectors via activation of local interneurons in IC, suggesting that the OFC controls extinction in a top-down manner through balancing activity of IC-NAc and IC-CeA projectors in local circuits.

To investigate the role of top-down inputs from OFC to IC (OFC → IC inputs) in memory processing, we optogenetically activated OFC axons in IC during fear conditioning and extinction (Fig. 7d, e and Supplementary Fig. 24a). Activation of OFC → IC inputs during fear conditioning did not affect the formation and expression of fear memory, whereas activation of OFC → IC inputs during the extinction markedly enhanced the efficacy of fear extinction (Fig. 7e), reminiscent of the effect induced by activation of IC-NAc projectors. These results indicate that OFC provides biased excitatory input towards IC-NAc projectors to selectively enhance extinction memory.

### Activation of the OFC → IC → NAc pathway promotes extinction learning and memory

To further delineate the role of the OFC → IC → NAc pathway in extinction memory, we exclusively expressed ChR2 in IC neurons receiving the top-down input from OFC and sending their axons to NAc, using an intersectional strategy combining AAV-mediated anterograde transsynaptic tagging[32] and retrograde infection from axonal terminals (Fig. 7f). We injected the anterograde transsynaptic AAV expressing Flp (AAV1-EF1α-Flp) into OFC, the Flp-dependent retrograde AAV expressing Cre (Retro-AAV-EF1α-fDIO-Cre-EGFP) into NAc, and the Cre-inducible AAV expressing ChR2 (AAV-DIO-ChR2-mCherry) in IC. This viral strategy enabled the Flp-induced expression of Cre specifically in IC neurons that receive the inputs from OFC and send axons to NAc, and further induced Cre-dependent expression of ChR2 in these IC neurons (Fig. 7f and Supplementary Fig. 24b). Optogenetically activating the axon terminals of these IC neurons only induced oEPSCs in NAc but not CeA neurons, confirming the labeling of IC neurons in OFC → IC → NAc pathway (Supplementary Fig. 25). Activation of IC neurons in the OFC → IC → NAc pathway during fear conditioning did not affect the formation and expression of the fear memory. In contrast, activating these neurons during extinction enhanced the efficacy of fear extinction (Fig. 7g). These results strengthen the notion that

the OFC → IC → NAc pathway is selectively engaged in extinction learning and memory.

## Discussion

The proper formation and expression of memory traces are critical for adapting to constantly changing environments and survival. However, how the brain organizes dedicated neuronal circuits that contribute to different constructs or ensembles of any given memory trace is still poorly understood. Using fear conditioning and extinction paradigm as a model, our work provides evidence that fear and extinction memories are respectively processed in IC by distinct subpopulations of projection neurons (IC-CeA and IC-NAc projectors). Interestingly, IC-CeA and IC-NAc projectors reciprocally inhibit each other via recruiting intracortical local interneurons, representing the neuronal mechanisms underlying the competition between fear and extinction memories. Furthermore, the top-down intercortical connections from OFC to IC preferentially activate IC-NAc projectors and thus facilitate extinction memory. These results provide mechanistic understanding of the neuronal circuits that control fear and extinction memory processing at intercortical, intracortical, and subcortical levels (Supplementary Fig. 26).

IC, mainly comprised of gustatory (GU) and agranular insular (AI) areas, plays important roles in supporting subjective feeling states by integrating sensory, emotional, and cognitive contents[11]. As a critical hub of cognition and emotion, IC has been shown to orchestrate appropriate behavioral responses to salient stimuli, including gating nociceptive hypersensitivity[33], guiding social communication[34], encoding aversive sensory states such as hunger, thirst, and anxiety[17,35,36], and controlling both aversive and appetitive tastant-reinforcement learning and memory[37–45]. Previous studies suggested that subregions of IC play different roles in various stages of fear memory[15–18]. Notably, a recent study revealed that IC integrates predictive sensory and interoceptive signals from vagus nerve to provide graded and bidirectional instructional signals that gate the extinction of learned fear[19]. However, it remains unknown whether subpopulations of IC neurons are responsible for distinct roles in the bidirectional regulation of fear memory. Here, we identified that distinct ensembles of IC projection neurons, mainly located in anterior IC, gate fear and extinction memories via functional connectivity to two subcortical regions, with fear-memory ensembles extensively innervating CeA and extinction-memory ensembles innervating NAc, respectively. Consistently, IC-CeA and IC-NAc projectors selectively contribute to fear and extinction memories, respectively. Thus, the differential modulation of fear memory by IC depends on the distinguished neuronal projections to CeA or NAc. The roles of IC-CeA and IC-NAc projectors in the integration of sensory and interoceptive signals remains to be investigated.

Furthermore, IC-CeA and IC-NAc projectors also contribute to the valence processing associated with fear and extinction memories. Activation of IC-CeA projectors induced conditioned aversion response in fear-experienced but not naïve mice, indicating that fear memory contributes to prime the activation of the IC-CeA projectors to aversive perception. Similarly, activation of IC-NAc projectors only established conditioned preference responses in mice that experienced fear extinction, consistent with the presumably appetitive nature of extinction memory[24,46,47]. These results suggest that the synaptic plasticity in long-range and local connections after memory acquisition play a key role in linking IC neuron activity with perception[48].

A fundamental question in the processing of fear and extinction memories is whether and how these two memories that led to opposite behaviors compete. Previous studies have shown that the medial prefrontal cortex, amygdala, and hippocampus are involved in the fear and extinction memories[3,8,9]. Despite some interactions, the fear and extinction ensembles are mainly considered independent of each other. Here we show that IC-CeA and IC-NAc projectors reciprocally

inhibit each other via recruiting local inhibitory neurons. The plasticity of this reciprocal inhibition is oppositely modulated by fear- and extinction-learning, and thus ensuring the proper expression of the dominant memory trace and suppression of the other one. Particularly, fear conditioning enhances the inhibitory inputs from fear-memory ensembles to extinction-memory ensembles, whereas extinction enhances the inhibitory inputs from extinction-memory ensembles to fear-memory ensembles. Thus, for the same auditory cue that predicts danger or safety before or after extinction, it activates distinct IC projectors, which in turn inhibit the other pathway to selectively express the appropriate behavior. Considering the diverse subtypes of cortical interneurons, the identity of interneurons involved in the competition between fear and extinction memories in IC awaits further investigations.

In addition, RV-assisted retrograde tracing also revealed selective connectivity between IC-CeA and IC-NAc projectors with neurons in subcortical areas. For example, CeA preferentially innervates IC-CeA projectors, while BLA prefers IC-NAc projectors. Since CeA is known to be involved in fear memory[20–23], whereas BLA contributes to both fear[49] and extinction memories[9], the competition between fear and extinction memories may also happen in subcortical areas before bottom-up propagation to cortical areas.

Top-down control plays important roles in sensory processing[50,51], perceptual learning[52], and memory adaptation[53], representing experience-dependent adaptation and integration of internal models for refined state estimation and goal-directed behavior. However, whether and how top-down intracortical projections control the fear and extinction memories remained as open questions[54]. Through the comprehensive retrograde mono-trans-synaptic tracing of IC projectors, we identified several differential neural inputs to IC-CeA and IC-NAc projectors. Notably, the OFC, a frontal cortical area associated with decision making, hedonic experience, and adaptive/flexible behaviors[55–58], was found to send more prominent projections to IC-NAc projectors than to IC-CeA projectors. This biased connection between OFC and different IC projectors was determined by slice electrophysiology. A potential technical limitation is that slicing the brain can potentially affect these intercortical connections. However, because of the close physical proximity of IC-CeA and IC-NAc projectors within the IC, it is unlikely that the synaptic connections identified here were unequally affected by the slicing procedure. Instead, it reflects the genuine difference between IC-CeA and IC-NAc projectors in vivo. Although additional in vivo measurements in the future are clearly needed, a recent study that OFC controls the response gain of the visual cortex and promotes visual-associative learning[59], implicated the executive role of OFC in regulating other cortical regions. Our experiment using intersectional genetic approach to selectively activate the OFC→IC→NAc circuit further demonstrated that this pathway increased the efficacy of extinction training, reinforcing the role of top-down control in extinction memory.

By exploring competitive and interactive memory ensembles of fear and extinction in IC, our study exemplifies the segregation of distinct neuronal memory ensembles in physical proximity (IC-CeA and IC-NAc projectors) to guide independent subcortical pathways but eliciting opposite behavioral outcomes associated with the same cue, following the top-down control from frontal cortices. Here, we propose a working model (Supplementary Fig. 26) to conceptualize the organization principles of different memory ensembles in which top-down connectivity may serve as a hierarchy gateway for memory processing along the continuum of intercortical, intracortical, and subcortical areas. Such generalized rules may also exist in the extensively studied tripartite neural circuit composed by the amygdala, prefrontal cortex, and hippocampus for fear and extinction[3,8,9], and may be further extended to other brain areas involved in memory ensembles in future studies. In view of this circuit hierarchy of memory differentiation and processing, our findings may inspire more

comprehensive identification of brain-wide long-range intercortical circuit wirings for memory ensembles, and inform strategies in the development of therapies to target specific pathways and molecular substrates against debilitating fear- and trauma-related memory disorders.

## Methods

### Mice

All animal procedures were approved by the Animal Ethics Committee of Shanghai Jiao Tong University School of Medicine and by the Institutional Animal Care and Use Committee (Department of Laboratory Animal Science, Shanghai Jiao Tong University School of Medicine; Policy Number DLAS-MP-ANIM. 01–05). All behavioral measurements were performed in awake, unrestrained, mice (male, 8–12 weeks old, C57BL/6J background). C57BL/6J mice were purchased from the Shanghai Laboratory Animal Center (SLAC) at the Chinese Academy of Sciences (Shanghai, China). Fos[2A-iCreER] (FosTRAP2) mice (stock no. 030323)[28] were purchased from Jackson Laboratory (Maine, USA). All mice were bred in specific pathogen-free laboratory animal facilities under standard conditions with temperatures of 21–23 °C, 40–60% humidity, and a 12-h light/dark cycle with rodent chow and water provided *ad libitum*. Adult male mice (6–12 weeks old) were used for various experiments. Experimental manipulations were performed during the light-on phase of the light/dark cycle, in accordance with the institutional guidelines.

### Fear conditioning and extinction

All auditory fear conditioning and extinction procedures were performed using the Ugo Basile Fear Conditioning System (UGO BASILE srl, Italy) according to a previous study[60]. Briefly, mice were first handled and habituated to the conditioning chamber for 3 successive days. The conditioning chambers (17 cm × 17 cm × 25 cm), equipped with stainless-steel shocking grids, were connected to a precision feedback current-regulated shocker (UGO BASILE srl, Italy). During fear conditioning, the chamber walls were covered with black-and-white checkered wallpapers, and the chambers were cleaned with 75% ethanol (context A). On day 1, mice were conditioned individually in context A with five pure tones (CS; 4 kHz, 76 dB, 20 s each) delivered at variable intervals (20–180 s), and each tone was co-terminated with a foot shock (US; 0.5 mA, 2 s each). ANY-maze software (version 6.31, Stoelting Co., USA) was used to automatically control the delivery of tones and foot shocks. Conditioned mice were returned to their home cages 60 s after the end of the last tone, and the floor and walls of the cage were cleaned with 75% ethanol for each mouse. For fear-memory retrieval, 24 h after conditioning, each mouse received four CS-alone presentations in a test chamber that had a gray non-shocking plexiglass floor and dark gray wallpaper, which was cleaned with 4% acetic acid solution between the tests for individual mice (context B). In order to minimize the expectation of mice to CS presentation, a different tone duration (i.e., 30 s) from that of conditioning (i.e., 20 s) was used in extinction and memory retrieval. For extinction training, mice trained in context A with five CS-US pairings on day 1 were presented with 12 CS presentations (4 kHz, 76 dB, 30 s each) without foot shock in context B on days 2–4. On day 5, for extinction retrieval, mice received four CS-alone presentations in the extinction context (context B). During behavioral testing, the chamber was placed in a sound-attenuating enclosure with a ventilation fan and a single house light (UGO BASILE srl, Italy). The movement of each mouse in the conditioning or test chamber was recorded using a near-infrared camera and analyzed in real-time via ANY-maze software. The ANY-maze behavior tracking software uses freezing score to represent the freezing status of the animal. The freezing score is a unit-less value as a result of rather complex calculations. When the software is calculating the freezing score, it looks for animal movements in the entire apparatus, which is accomplished by comparing every pixel of the current

frame to earlier ones. If the software fails to find any movement (large number of flickering pixels) in the apparatus, the animal would be considered to be freezing. The software also includes 'noises' of the video when calculating the freezing score, for example, a breathing animal would cause some pixels to flicker. These kinds of noises coming with animals' physiological activities would have an influence on the value of the freezing score. Typically, louder video noise would result in lower freezing score of the animal at that frame. Finally, the software would give a result of freezing score at each frame, and periods that the animal is considered to be freezing according to the threshold setting. A fear response was operationally defined as measurable behavioral freezing (more than 1-s cessation of movement), which was automatically scored and analyzed by ANY-maze software. For animals connected to an optical fiber to the head, light stimuli during test sessions can interfere with the program's motion detection, so freezing of these sessions was scored independently for each video by an experienced experimenter in a double-blind manner. The time spent freezing during the tone (cue) was measured for each tone presentation.

## Viral constructs

The following viruses were purchased from Obio Technology Co. Ltd. (Shanghai, China) and were used in the present study: Retro-AAV-Syn-EGFP (Serotype 2/retro), Retro-AAV-Syn-mCherry (Serotype 2/retro), Retro-AAV-Syn-Cre-EGFP (Serotype 2/retro), AAV1-EF1a-Flp (Serotype 2/1), AAV-EF1α-fDIO-Cre-EGFP (Serotype 2/retro), AAV-EF1α-DIO-ChR2-E123T/T159C-mCherry (Serotype 2/9), AAV-EF1α-DIO-mCherry (Serotype 2/9), AAV-EF1α-DIO-eNpHR-EYFP (Serotype 2/9), and AAV-EF1α-DIO-EYFP (Serotype 2/9). AAV-CAG-DIO-TetTox-EGFP (Serotype 2/9) and AAV-CAG-DIO-EGFP (Serotype 2/9) were packaged by Shanghai SunBio Biomedical Technology Co. Ltd. (Shanghai, China). Retro-AAV-Syn-Cre (Serotype 2/retro), AAV-EF1α-DIO-RVG (Serotype 2/9), AAV-EF1α-DIO-His-EGFP-TVA (Serotype 2/9), and RV-ENVA-dG-DsRed (titer $3.1 \times 10^8$ IFU/ml) were purchased from Brain VTA (Wuhan, China). AAV-Syn-DIO-GCaMP6m (Serotype 2/9) was produced by Shanghai Taitool Bioscience Co. Ltd. (Shanghai, China). All viral vectors were stored in aliquots at −80 °C until further use. Except where otherwise indicated, all the viral titers of AAVs for injection were more than $10^{12}$ viral particles per ml.

## Viral injections

Mice at 6−7 weeks old were anesthetized with 1% sodium pentobarbital via a single intraperitoneal injection (10 ml per kg of body weight), after which each mouse was mounted in a stereotactic frame with non-rupture ear bars (RWD Life Science, Shenzhen, China). After making an incision to the midline of the scalp, small bilateral craniotomies were performed using a microdrill with 0.5-mm burrs. Glass pipettes (tip diameter: 10−20 μm) were made with a P-97 Micropipette Puller (Sutter glass pipettes, Sutter Instrument Company, USA) for AAV microinjections. The microinjection pipettes were first filled with silicone oil and were then connected to a microinjector pump (RWD Life Science, Shenzhen, China) to achieve full air exclusion. AAV-containing solutions were loaded into the tips of pipettes and injected at the following coordinates (anteroposterior to bregma, AP; lateral to the midline, ML; below the bregma, DV; in mm): IC: AP, +0.98, ML, ±3.30, DV, −3.80; NAc: AP, +1.42, ML, ±1.12, DV, −4.25; CeA: AP, −1.22, ML, ±2.65, DV −4.80; and OFC: AP, +2.86, ML, ±1.00, DV, −2.50. Virus-containing solutions were injected bilaterally/unilaterally into the IC (0.5 μl/side), NAc (0.3 μl/side), CeA (0.2 μl/side), or OFC (0.3 μl/side) at a rate of 0.1 μl/min. After injection, the pipette was left in place for an additional 10 min to allow the injectant to diffuse adequately. Mice were allowed to recover for at least four weeks before behavioral and other tests, and the injection sites were examined at the end of the experiment by the expression of the fluorescent proteins, EGFP, EYFP, or mCherry.

## Optogenetic manipulations during fear learning or extinction training

To investigate optogenetic-mediated effects during either fear learning or extinction training, mice were implanted with light-emitting diode (LED) optical connectors (200 μm O.D., N.A. = 0.37, λ = 470 nm) into the IC at two weeks after viral injections. For optogenetic activation, an external receiving end was used to connect a 470-nm wireless optogenetic system (Hangzhou Newdoon Technology Co. Ltd, Hangzhou, China) to the implanted LED optical connectors in each mouse. The light pulses were controlled in coordination with the fear conditioning system (UGO BASILE srl, Italy), where blue light (470 nm, 4−6 mW) was delivered in 10-ms pulses at 20 Hz during either fear conditioning (duration: 30 s) or extinction training (duration: 40 s). The duration of light pulses exceeded 5 s both before and after the CS to ensure that covered the entire CS-US pairing for fear conditioning (20-s duration for each tone) or full CS exposure for extinction training (30-s duration for each tone). For optogenetic inhibition, a wired optogenetic system was used. An external optical fiber was used to connect a 590-nm laser power source (Changchun New Industries Optoelectronics Technology Co., Ltd., China) to the implanted optical fiber (200 mm O.D., N.A. = 0.37) in each mouse. The external optical fiber was attached to a rotary joint (FRJ_1 × 1_FC-FC, Doric Lenses, Canada) to allow the mouse to freely behave. Each test mouse was allowed to habituate in its home cage with the external fiber attached for at least 10 min. Laser pulses were controlled through a customized MATLAB program (AniLab Software and Instruments, Shanghai, China). For optogenetic inhibition, yellow light (590 nm) was delivered in a continuous pattern during presentation of each CS-US paring for 30 s (during fear conditioning) or 40 s (during extinction training), with the final output power ranging from 8 to 10 mW depending on the light transmission efficacy of the optical fiber used.

## Fiber photometry

Fiber photometry experiments were performed as previously described with modifications[61] using a fiber photometry system (Thinker Tech, Nanjing, China). Fluorescent signals produced by an excitation laser beam from a 488-nm laser (OBIS 488LS; Coherent) was reflected by a dichroic mirror (MD498; Thorlabs), focused by a 10× objective lens (NA = 0.3; Olympus) and coupled to a rotary joint (FRJ_1 × 1_FC-FC, Doric Lenses, Canada). Two weeks after the AAV injection, an optical fiber (200 mm O.D., NA = 0.37) was implanted into the IC as described above. The laser power was adjusted at the tip of the optical fiber to 40−60 μW to minimize bleaching of the GCaMP6m probes. Excitation fluorescence was collected by the same multi-mode optical fiber and was converted into electrical signals by two low-light detectors at the detection end to reflect different neural-activity information. The analog voltage signals were digitalized at 100 Hz using a Power 1401 digitizer and Spike2 software (CED, Cambridge, UK). During the behavioral tests−including fear learning, extinction training, or fear or extinction memory retrieval−GCaMP6m fluorescent intensities were recorded. The pre-sound signal was set as the baseline.

The averaged $Ca^{2+}$ response was calculated via MATLAB. Photometry data were exported to MATLAB mat files from Spike2 for further analysis. Data were segmented based on behavioral events within individual trials. Fluorescent changes (ΔF/F) were calculated by (F − $F_0$)/$F_0$, where $F_0$ refers to the median of the fluorescence values during the baseline period (from the 10-s preceding onset of each CS). The ΔF/F values were presented as heatmaps or per-event plots, with shaded areas indicating the standard error of the mean (SEM). Permutational tests were used to analyze the statistical significance of event-related fluorescent changes, as previously described. To statistically quantify the change in fluorescent values, the peak ΔF/F was defined as the maximal fluorescent changes during both baseline (the 10-s control time window before onset of the first CS) and event (maximal values detected within 20 s after the tone or shock onset during the

conditioning phase or maximal values detected within 30 s after the tone onset during fear or extinction memory retrieval) periods. Moreover, the area under the curve (AUC, $\Delta F/F \times s$) was calculated from the same set of data. In order to analyze the calcium dynamics during the event, the peak $\Delta F/F$ and AUC of the tone responses for the first 5-s of the CS and the 2-s shock period were also quantified. In addition, the data as z-scores were also reported to reduce the standard deviation of the signal in baseline before each CS.

## Neuronal tagging

Neuronal tagging via FosTRAP strategy was performed according to previous studies[27,28,62,63], with modifications. Activity-dependent recombination was induced with 4-hydroxytamoxifen (4-OHT, Sigma-Aldrich, Catalog no. H6278, USA). In brief, 4-OHT was dissolved at 20 mg/ml in ethanol by shaking at 37 °C for 15 min and was then aliquoted and stored at −20 °C for up to several weeks. Before use, 4-OHT was redissolved in ethanol by shaking at 37 °C for 15 min, corn oil (Sigma-Aldrich, Catalog no. C8267, USA) was added to yield a final concentration of 10 mg/ml of 4-OHT, and the ethanol was evaporated by vacuum under centrifugation. The final 10 mg/ml of 4-OHT solutions were stored for, at most, 24 h at 4 °C before use. All injections were delivered intraperitoneally. Mice were transported from the vivarium to an adjacent holding room at least 3 h before the 4-OHT injections to minimize transportation-induced immediate early gene activity. For fear memory, activity-dependent neuronal tagging was induced by a single intraperitoneal injection of 4-OHT (20 mg/kg mice) at 30 min prior to fear conditioning, whereas the control group (i.e., CS only) was induced prior to a similar behavioral treatment, but without footshock. For extinction memory, activity-dependent neuronal tagging was induced by a single intraperitoneal injection of 4-OHT (20 mg/kg mice) at 30 min after 2 consecutive days of extinction training, whereas the control group (i.e., Ext. CS only) was induced after a similar behavioral treatment, but without footshock during fear conditioning. Mice were then returned to the vivarium with a regular 12-h light/dark cycle for the remainder of the experiment. The effect of reactivating fear- or extinction-memory-associated IC neurons on the retrieval efficiency of corresponding memories was measured quantitatively, as previously described[62].

## Conditioned place aversion or preference assay

To test the roles of IC-CeA projectors and IC-NAc projectors in aversive and appetitive valence processing, we developed light-related conditioned place aversion (CPA) and conditioned place preference (CPP) protocols based on a previous publication[61]. The light-related CPA assay consists of three phases: pre-test, acquisition, and post-test. All phases were conducted under red light and sound-attenuated conditions. For naïve mice, a two-compartment place preference apparatus was used. It consists of two unique conditioning chambers that allow for unbiased entry into either chamber. During the pre-test phase (day 1), the mouse was placed in a randomly selected chamber and allowed to freely explore the entire apparatus for 15 min. Animals with significant bias towards either chamber were excluded from subsequent experiments. The acquisition phase consisted of 3 successive days with two conditioning trials (morning versus afternoon) each day for a total of six acquisition trials. In the morning session, the mouse was limited in one conditioning chamber (light-paired, learning chamber), where it received blue light (473 nm) in 5-ms pulses at 20 Hz, with the 5-min light-on period following a 5-min light-off period, repeated 3 times over a total period of 30 min. In the afternoon session, the same mouse was restricted to the opposite conditioning chamber, where it received no-light stimulation. On day 3, the mouse stayed without light in the morning and with light in the afternoon. On day 4, it was conditioned with light in the morning and without light in the afternoon. On the test day (day 5), all mice were placed randomly in a chamber and allowed to freely explore both chambers for 15 min. The time the animal spent in

each chamber was analyzed to identify a CPA (or CPP) to the no-light- or light-paired chamber. For mice that had acquired fear or extinction memories, a three-compartment place preference apparatus was used. It consists of two unique conditioning chambers with a neutral middle chamber that allows for unbiased entry into either conditioning chamber. During the pre-test phase (day 1), mice were placed into the middle chamber and allowed to freely explore the entire apparatus for 15 min. The acquisition phase is as described above. On the test day (day 5), all mice were placed in the middle chamber of the apparatus and allowed to freely explore all three chambers for 15 min. The time the animal spent in each chamber was analyzed to identify a CPA (or CPP) to the no-light- or light-paired chamber.

All behavioral sessions were video recorded using Noldus Etho-Vision XT (version 16.0, Noldus Information Technology, Netherlands). Behavioral analysis of the CPA data was performed by assessing (1) normalized CPA score (post-test duration spent in the light-paired chamber divided by the pre-test duration spent in the same chamber), (2) preference for light-paired chamber (duration spent in the light-paired chamber divided by that in the light-unpaired chamber for the post-test), and (3) times (in sec) spent in light-paired chamber during pre-test and post-test for individual animals. Behavioral analysis of the CPP data was performed by assessing (1) CPP score (post-test duration spent in the light-paired chamber divided by the pre-test duration spent in that chamber), (2) preference for conditioned chamber (duration spent in the light-paired chamber divided by that in the light-unpaired chamber for the post-test), and (3) times (in sec) spent in light-paired chamber during pre-test and post-test for individual animals.

## Slice electrophysiology

Whole-cell recordings were performed in acute brain slices from behaviorally trained mice[60] and/or those that had been stereotaxically injected with AAVs, as indicated in different figures. Preparation of the brain slices used for electrophysiological recordings was performed by an investigator with knowledge of the identity of the experimental group, while the collection and analysis of electrophysiological data were done independently by an experienced experimenter in a blind manner. Mice were deeply anesthetized with 1% sodium pentobarbital and were subsequently decapitated. Brains were dissected quickly and were chilled in well-oxygenated (95% $O_2$/5% $CO_2$, v/v) ice-cold artificial cerebrospinal fluid (aCSF) containing the following (in mM): 125 NaCl, 2.5 KCl, 12.5 D-glucose, 1 $MgCl_2$, 2 $CaCl_2$, 1.25 $NaH_2PO_4$, and 25 $NaHCO_3$ (pH 7.35–7.45). Coronal brain slices (300-μm thick) containing regions of the IC, CeA, or NAc were cut with a vibratome (Leica VT1000S, Germany). After recovery for 1 h in oxygenated aCSF at 30 ± 1 °C, each slice was transferred to a recording chamber and was continuously superfused with oxygenated aCSF at a rate of 1–2 ml per min. The principal neurons in the IC, or the neurons in the CeA or NAc, were patched under visual guidance using infrared differential-interference contrast microscopy (BX51WI, Olympus, Japan) and an optiMOS camera (QImaging, Teledyne Imaging Group, Canada). The slices were continuously perfused with well-oxygenated aCSF at 35 ± 1 °C during all electrophysiological experiments. Whole-cell patch-clamp recordings were performed using an Axon 200B amplifier (Molecular Devices, USA). Membranous currents were sampled and analyzed using a Digidata 1440 interface and a personal computer running Clampex and Clampfit software (pCLAMP 10.5, Molecular Devices, USA). Access resistance was 15–30 MΩ, and only cells with a change in access resistance <20% were included in the analysis. Optical stimulation of ChR2- or eNpHR-expressing neurons was performed using a collimated LED (Lumen Dynamics Group Inc, USA) with peak wavelengths of 470 or 590 nm, respectively. The LED was connected to an Axon 200B amplifier to trigger photostimulation. The brain slice in the recording chamber was illuminated through a 40× water-immersion objective lens (LUMPLFLN 40XW, Olympus, Japan). The intensity of

photostimulation was directly controlled by the stimulator (2–18 mW/mm$^2$), while the duration was set through Digidata 1440 and pClamp 10.5 software. The functional potency of the ChR2-expressing virus was validated by measuring the numbers of APs elicited with different frequencies of blue-light stimulation (1 ms; 5, 10, and 20 Hz) and the inward photocurrents (500-ms pulse) mediated by ChR2 in brain slices. To corroborate the functional potency of eNpHR-mediated optogenetic inhibition, yellow light (λ = 590 nm, 500-ms pulse) was delivered to reduce spikes to current injection under current-clamp mode.

**Light-evoked EPSCs.** To evoke synaptic responses in the NAc or CeA by optogenetic photostimulation of IC axons or those in the IC-CeA or IC-NAc projector by optogenetic photostimulation of OFC axons or axons of IC-NAc or IC-CeA projector, each slice was illuminated every 20 s with blue-light pulses of 5-ms duration. To prevent polysynaptic activities from being detected in EPSC recordings, we applied appropriate photostimulation intensities that produced 30–50% of the maximal synaptic response. Optical activation of ChR2-expressing axons was performed using a blue collimated LED with a peak wavelength of 470 nm (Lumen Dynamics Group Inc, USA). The LED was connected to an Axon 200B amplifier to trigger photostimulation. The brain slice in the recording chamber was illuminated through a 40 × water-immersion objective lens (Olympus LUMPLFLN 40XW, Japan). The intensity of photostimulation was directly controlled by the stimulator, while the duration was set through Digidata 1440 and pClamp 10.5 software. For recording the light-evoked EPSCs, the recording pipettes (3–5 MΩ) were filled with a solution containing the following (in mM): 132.5 cesium gluconate, 17.5 CsCl, 2 MgCl$_2$, 0.5 EGTA, 10 HEPES, 4 Mg-ATP, and 5 QX-314 chloride (280–300 mOsm, pH 7.2 with CsOH). To determine the paired-pulse ratio (PPR), neurons were voltage clamped at −70 mV. The AMPAR oEPSCs were evoked by paired photostimulation (50-ms interval; 5-ms duration) of ChR2-expressing axons, and the PPR was calculated as the peak amplitude ratio of the second-to-the-first oEPSC. To calculate the IPSC/EPSC ratio, the recorded neurons were clamped at −40 mV, and the recording pipettes (3–5 MΩ) were filled with a solution containing the following (in mM): 145 potassium gluconate, 5 NaCl, 10 HEPES, 2 MgATP, 0.1 Na$_3$GTP, 0.2 EGTA, and 1 MgCl$_2$ (280–300 mOsm, pH 7.2 with KOH). The inward current was designated as the EPSC while the outward current was designated as the ISPC.

**Spike firing.** Spiking activity and related membranous properties of different populations of IC neurons were measured with an internal solution containing the following (in mM): 145 potassium gluconate, 5 NaCl, 10 HEPES, 2 MgATP, 0.1 Na$_3$GTP, 0.2 EGTA, and 1 MgCl$_2$ (280–300 mOsm, pH 7.2 with KOH). Data were analyzed by the Mini-iAnalysis Program (Version 6.0.1, Synaptosoft, USA) with an amplitude threshold of 20 mV.

**Rabies-virus-based retrograde monosynaptic tracing**
Rabies-virus-based retrograde monosynaptic tracing was performed according to a previous study[64]. Firstly, 100 nl of Retro-AAV-Syn-Cre was injected into the NAc or CeA, and 100 nl of a 1:1 volume mixture of AAV-EF1α-DIO-RVG and AAV-EF1a-DIO-EGFP-T2A-TVA was injected into the IC in the same mouse, such that rabies glycoprotein (RVG) and TVA receptor were expressed in the IC-CeA or IC-NAc projectors. After three weeks, 200 nl of glycoprotein-deleted (ΔG) EnvA-pseudotyped rabies-virus RV-ENVA-ΔG-DsRed was injected into the IC to enable retrograde monosynaptic tracing from different populations of IC neurons. Two weeks after rabies-virus injection, mice were sacrificed and perfused. After paraformaldehyde fixation and sucrose dehydration, consecutive 50-μm coronal sections of the whole brain were prepared on a cryostat (CM1900, Leica, Japan). After tissue sectioning, immunofluorescent staining was used to confirm the virus injection

site by the observed colocalization of red and green fluorescence in the IC. The sections were imaged using an Olympus VS120 virtual microscopy (Olympus, Japan) slide-scanning system. Cell counting was performed manually using ImageJ software (NIH Image, version 1.80, USA). A previous software package developed in Matlab (version R2014a, USA) was used to analyze the digitized brain images[64]. This analysis software consists of three modules: image registration, signal detection, and quantification/visualization. The data were presented as the number of labeled neurons in a given region divided by the total number of labeled neurons detected in the whole brain (excluding the injection site as the starter cells). The figure was displayed by aligning fluorescence imaging to the corresponding coronal section of Allen Mouse Brain Atlas and Allen Reference Atlas - Mouse Brain. Allen Mouse Brain Atlas, mouse.brain-map.org and atlas.brain-map.org.

**Histology and fluorescent immunostaining**
Mice were deeply anesthetized with 1% sodium pentobarbital and were transcardially perfused with saline followed by ice-cold PBS. The brains were then dissected and immersed in 4% paraformaldehyde in PBS. Coronal brain slices containing the whole IC, NAc, CeA, and/or OFC were sectioned (30-μm or 50-μm thicknesses) using a vibratome (VT1000S, Leica, Japan) and were processed for *post-hoc* analysis of viral infection efficiency and specificity. After a 15-min incubation in 4,6-Diamidino-2-phenylindole dihydrochloride hydrate (DAPI) solution (1: 2000), sections were washed four times (15 min each time) in PBS with 0.1% Tween-20. Slides were mounted in the dark with glass coverslips using mounting media. Stained slides were prepared for microscopy. For c-fos or Cre recombinase immunofluorescent staining, slices were blocked with permeable buffer (0.3% Triton X-100 in PBS) containing 10% donkey serum for 1 h at room temperature and were incubated with rabbit anti-c-fos primary antibody (1:500; Cell Signaling Technology, catalog no. 2250, USA) or rabbit anti-Cre primary antibody (1:500; catalog no. 15036, Cell Signaling Technology, USA) in permeable buffer containing 2% donkey serum overnight at 4 °C. The slices were then washed four times with PBST (0.1% Tween-20 in PBS) for 15 min each and were then incubated with donkey anti-rabbit IgG (H + L) Alexa Fluor secondary antibodies (1:200; Alexa Fluor™ 488, catalog no. A21206, ThermoFisher Scientific, USA; or Alexa Fluor™ 647, catalog no. A31573, ThermoFisher Scientific, USA) and DAPI (1:5000; catalog no. D1306, ThermoFisher Scientific, USA) in PBS buffer for 2 h at room temperature. Slices were washed four times in PBS-T and were mounted on glass slides using the mounting media. The fluorescent signals were imaged using confocal microscopes including Leica (TCS SP8, Japan), ZEISE (LSM 710, Germany), and/or Nikon (Digital Eclipse A1R+, Japan). For quantification of immunofluorescent labeling, ImageJ software (NIH Image, version 1.8.0, USA) was used to manually count the fluorescence-positive cells and the fluorescence of IC projections. Moreover, by using standard histological methods and confocal microscopy, the locations of optical fiber tips were also validated for all behavioral experiments.

**Statistics and reproducibility**
All data presented in this work were obtained from at least three biological replicates independently; that is, multiple animal cohorts from different litters, at least three experimental repeats for each micrograph. All attempts of replication were successful. The graphs were created by the Origin Software (version 9.5, OriginLab Corporation, Northampton, MA). Data are presented as the mean ± the SEM unless indicated otherwise. Most histograms display individual data points that represent the values and numbers of individual samples for each condition. Data distributions were tested for normality, and homogeneity of variance among groups was assessed using the Levene's test. Statistical comparisons were performed using two-tailed Student's *t* test as well as one-way analyses of variance (ANOVAs) or two-way repeated measures ANOVAs. For post hoc analysis, we used

Bonferroni's corrections for multiple comparisons. Statistical analysis was performed with GraphPad Prism (version 8.0.2, GraphPad Software, Inc., USA) or Office 2019 (Microsoft, USA), and $P < 0.05$ was considered statistically significant. Significance is mainly displayed as $^*P < 0.05$, $^{**}P < 0.01$, and $^{***}P < 0.001$; however, in some cases it is indicated as $^\#P < 0.05$, $^{\#\#}P < 0.01$, and $^{\#\#\#}P < 0.001$ for multiple comparisons; N.S. denotes non-significant values.

## Reporting summary

Further information on research design is available in the Nature Research Reporting Summary linked to this article.

## Data availability

All data supporting the findings described in the paper are available in the article and in the Supplementary Information. Source data are provided with this paper.

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

## Acknowledgements

This study was supported by grants from the Ministry of Science and Technology China Brain Initiative Project (2021ZD0202800), the National Natural Science Foundation of China (31930050, 32071023, 31900701, and 81903583), the Science and Technology Commission of Shanghai Municipality (18JC1420302 and 22XD1420700), the Shanghai Municipal Science and Technology Major Project (2018SHZDZX05), the Shanghai Jiao Tong University College of Basic Medical Sciences (YCTSQN2021002), and innovative research team of high-level local universities in Shanghai.

## Author contributions

Qi Wang, W.-G.L., and T.-L.X. conceived the project, designed the experiments, and interpreted the results. Qi Wang and J.-J.Z. performed the majority of behavioral experiments, animal surgery, immunohistochemistry, and data analysis. Y.-P.K., Y.-J.W., X.G., X.Y., Z.-J.L., Qin Wang, Q.J., and Y.L. assisted with some of the behavioral experiments and conducted viral injections. Qi Wang and W.-G.L. performed slice recording and data analysis. Qi Wang, L.W., Y.-M.L., and S.Z. performed rabies-virus-dependent retrograde tracing and data analysis. J.-F.L., M.-G.L., N.-J.X., M.X.Z., L.-Y.W., and S.Z. contributed to data interpretation and experimental design. W.-G.L., S.Z., M.X.Z., L.-Y.W., and T.-L.X. wrote the paper with contributions from all authors. All authors read and approved the final paper.

## Competing interests
