## [Peer Review File · Nature Communications]

Insular Cortical Circuits as an Executive Gateway to Decipher Threat or Extinction Memory via Distinct Subcortical PathwaysREVIEWER COMMENTS

Reviewer #1 (Remarks to the Author):

In this manuscript, Wang and colleagues investigate the roles of two distinct projection-defined insular cortex neuron subpopulations in fear conditioning and extinction. Major findings of this study include: i) Fear memory- and extinction memory-encoding neurons are found to be enriched in the anterior insular cortex (aIC) using the FosTRAP method. Furthermore, optogenetic inhibition of fear- and extinction-memory ensembles in aIC impaired memory retrieval for fear and extinction memories, respectively. ii) fear and extinction memories are segregated into distinct and non-overlapping IC projection neurons with central amygdala (CeA) and nucleus accumbens (NAc) projectors encoding fear and extinction memories, respectively. iii) IC-CeA and IC-NAc projectors reciprocally inhibit each other through intracortical interneurons. iv) whole-brain mapping of monosynaptic inputs to IC-CeA and IC-NAc projecting IC neurons reveal that IC-NAc projectors receive significantly more input from the orbitofrontal cortex (OFC). Furthermore, intercortical connectivity between OFC and IC projectors gate extinction learning and memory. v) Finally, activation of OFC to IC to NAc pathway promotes extinction learning and memory. The results of these experiments are convincing and provide important insights into the neural circuit mechanisms underlying fear and extinction memories.

Previous studies have demonstrated that IC-CeA and IC-NAc projection neurons, particularly ones located in the posterior IC, play an important role in aversive state processing (Gehrlach et al., *Nature Neuroscience*, 2019). Furthermore, a recent publication further showed that IC acts as a state-dependent regulator of fear which is necessary to set equilibrium between extinction and fear memories (Klein et al., *Science*, 2021). The present study complements and extends these previous findings significantly by investigating the role of projection-defined IC neurons in the anterior IC in fear and extinction learning. Overall, the present study is timely and makes an important contribution to the fear and extinction literature. However, there are points that need to be addressed.

Major points:

1) As mentioned above, in a paper published a few weeks ago Klein and colleagues demonstrate the role of IC in establishing the balance between fear and extinction memories. It is necessary to cite the Klein et al., *Science*, 2021 paper and discuss the findings of the present study in light of Klein et al. findings, particularly emphasizing the differences and the overlap between the two studies.

2) There are several issues regarding the fiber photometry experiments in Figure 3:

i) The authors state that in calculating dF/F , F_0 refers to the median of fluorescence values during the baseline period which they took as the 5-s period before the onset of the first CS. However, the behavior of animals and their fear state changes with each CS through the course of fear conditioning and also the course of fear retrieval and extinction retrieval sessions. As a result, neuronal activity is expected to change through the course of these sessions as well. Therefore, 5-s period before the first CS would not reflect the calcium transients and fluorescence levels during the pre-CS period of each CS. Photobleaching during a session is also expected to affect baseline fluorescence levels through the course of these sessions. It is therefore more appropriate to use the baseline period for each CS (5-s before each CS) separately and analyze the dF/F for each CS by using the F_0 value calculated by using the 5-s period before each CS. Please re-analyze the dF/F values in Figure 3 accordingly.

ii) In Figure 3e and 3g, it would be good to see how the responses are after the tone offset. Please include at least 10-s post-tone period in these graphs. It would also be good to see a longer baseline period (e.g. 10-s) before the tone.

iii) In Figure 3g, the fluorescence level during baseline changes from fear to extinction retrieval likely reflecting the change in fear state and movement level of the animals. In the heat maps, baseline activity seems to have high transients in most animals during extinction retrieval. For instance, in animal #1, there is increased activity during the CS however the activity is quite high before the CS, as well. It would therefore be good to see a longer baseline period before the CS, as I also stated in my previous comment. Also, since the variability in neural activity during baseline seems to be high in extinction retrieval, it would be good to report the data as z-scores which will take into consideration the standard deviation of the signal in baseline before each CS.

iv) In Figures 3d and 3f, responses to the tone seem to increase particularly during the onset of the tone and these tone onset responses seem to increase for both IC-CeA and IC-NAC projectors through the course of fear conditioning. Analyzing the tone responses for the first 5-s (rather than the whole duration) of the CS might yield significant increases from the first to the last tone. How do the middle figures showing AUC values in Fig. 3d and 3f look if first 5-s of CS is analyzed? If there is a significant increase from first to last tone for both IC-CeA and IC-NAC projectors, the authors will need to reinterpret these findings, particularly for the IC-NAC projector group. Also, how do Figures 3e and 3g look if AUC is calculated for the first 5 sec of the tone?

v) In Figures 3d and 3f, responses to shock are analyzed by calculating the AUC during the 20-s period (2-s shock and 18-s post-shock) after the tone. However, the shock is only 2-s long and it is not clear why the authors included such a long post-shock period into their analysis. To see responses to the shock, the authors should rather calculate the AUC values during the 2s shock period. How do the right figures in Figures 3d and 3f look if analysis is restricted to 2s?

3) In Figures 3, 4 and supplementary Figure 5, the freezing levels are quite different when the two projector groups are compared. For instance, in Figure 3c, mice in IC-CeA group freeze much less (~ 60%) compared to IC-NAC group (~80%) at the end of fear conditioning. Also, when we compare the control mice in Figure 4f versus 4g, the control mice in IC-CeA projector group (Fig. 4f) freeze significantly lower compared to the control mice in IC-NAC projector group (Fig. 4g). The fear conditioning parameters

seem to be same between these two groups, what can be the reason for this discrepancy in freezing levels between the groups? Please discuss the reasons for this discrepancy?

4) In Figure 1, there is inter-individual variability in freezing levels of mice with some mice freezing more and some less during fear retrieval (Fig. 1e) and extinction retrieval (Fig. 1j) tests. It would be interesting to see and also will strengthen the findings of this experiment if there is a correlation between freezing levels and the number of FosTRAPed cells. For instance, one would expect that the more number of FosTRAPed cells in an animal will correlate with higher freezing levels during fear retrieval and vice versa for extinction retrieval.

5) In Figures 3d and 3f (middle and right), the authors state that they performed one-way ANOVA. Did they perform repeated measures? If not, repeated measures ANOVA should be done.

6) In Supplementary Figure 13, the authors find that OFC inputs to IC-CeA projectors provided more inhibition than excitation indicating a net effect of inhibition of this input on these IC neurons. Therefore, one would expect that if OFC input to IC is activated during fear conditioning, this should lead to impaired fear learning and retrieval since IC-CeA neurons are inhibited. However, optogenetic activation of OFC input to IC in Figure 6e does not have any effect on fear acquisition and retrieval contrary to expectation. Can the authors discuss possible reasons for this.

Minor points:

7) The authors should include histological verification of optic fiber placements and virus expression profiles of all mice in each of the experiments as supplementary figures.

Reviewer #2 (Remarks to the Author):

In this study, the authors characterized how the insular cortex (IC) mediates fear expression and extinction through projections to the central amygdala (CeA) and the nucleus accumbens (NAc), respectively. Using state-of-the-art engram tagging, optogenetics, chemogenetics, viral intersectional approaches, photometry, slice electrophysiology and tracing, the authors found how IC-CeA and IC-NAc circuits were necessary and sufficient for fear and extinction memories, respectively. Moreover, the authors found that IC-CeA and IC-NAc neurons gated each other's activity through local feedforward inhibition, and that the orbitofrontal cortex promoted fear extinction through projections to the IC.

This is an outstanding, innovative and creative study that will have a positive impact in the Pavlovian fear field. The approach was straightforward, the experiments were comprehensive and the narrative was clear and concise. I have identified several issues that I hope the authors can address before moving forward with this manuscript.

Major issues:

- 1) The study lacks histology reports throughout, aside from single subject examples. The authors need to report virus spread, and optic fiber and placements for all corresponding experiments.
- 2) The authors should report how their optogenetic and chemogenetic, or ablation manipulations affect general locomotion.

Minor issues

- 1) The authors should show the fear conditioning freezing curves in Figure 1. It is not clear why the authors tagged neurons after two extinction sessions, as this would include neurons involved in fear retrieval (early extinction session), extinction retrieval (because they already extinguished the previous day) and further extinction learning. Reporting the freezing curves will allow the reader to better interpret the results in Figure 1.
- 2) In Figure 1, when the authors report extinction cell tagging, they reported a CS only control, but not a fear reactivation control group. This is important because fear extinction retrieval involves both fear and extinction retrievals. It is unclear if the tagged cells account for fear or extinction reactivation because we do not know how many cells would have been recruited by fear retrieval on its own.
- 3) It would be good to see the mean fluorescence of IC-CeA and IC-NAC ensembles outside of the fluorescence ratio as well.
- 4) Photometry experiments lack a within subject control (mCherry or isosbestic channel) to account for movement artifacts.
- 5) The authors should be cautious when describing effects on fear or extinction retrieval after employing manipulations that affected fear or extinction acquisition. To clearly see effects on expression, the authors need to limit their manipulations to the expression test period and avoid confounds of acquisition effects.
- 6) The authors claim that there was a significant increase in CPP score in Line 281, but Figure 8u depicts a p value of 0.09, which did not reach significance.
- 7) What is the rationale of having different tone durations for conditioning, extinction and retrieval tests?
- 8) The authors claim that activation of OFC-IC projections enhanced fear extinction without affecting fear conditioning (Figure 6e). We know from Extended Data Figure 7b that stimulation of IC-CeA does

not further enhance fear conditioning when using a high shock. Thus, it is possible that using a high shock in Figure 6e could have mask an effect of fear enhancement from stimulating OFC-IC projections.

9) The authors report that 100% of IC-NAc projectors receive direct excitatory inputs from OFC, whereas IC-CeA only received 69%, suggesting that OFC projections to IC are biased towards NAc projectors (Extended Data Figure 13c). It is unclear whether the authors can conclude this with only 18 and 16 neurons per projection. The authors should employ a statistical test, such as a Pearson's, to corroborate their claim.

10) Is Extended Data Figure 10a group data? If so, the authors should report the error bars.

Reviewer #3 (Remarks to the Author):

In the present study, the authors made an excellent and complete job on studying the role of neuronal projections coming from the Insular cortex (IC) to either the Central Amygdala (CeA) or to the Nucleus Accumbens (NAc) during fear conditioning and extinction phases (learning and memory). The authors show that while IC-CeA projection is critical for fear conditioning (and its retrieval/expression), the IC-NAc projection is important for extinction learning and its retrieval. Interestingly, they also showed that when one of these projections is active, the other projection is indirectly inhibited (reciprocal inhibition between IC neurons projecting to either CeA or to NAc, perhaps via interneurons). Lastly, the authors dissected the afferent inputs coming from different brain regions to IC neurons projecting to CeA or to NAc. Then, they subsequently focused on the OFC contribution in regulating fear conditioning and extinction phases. The authors particularly showed that activating the OFC-IC-NAc pathway facilitated extinction learning and its memory (retrieval). Overall, this is a very compelling and important set of findings that complement previous data in the field. I do have several suggestions/comments to be addressed in a revision.

Comments

1. The results section is hard to follow (disorganized). Please check the sequence/flow of the main and supplementary figures in the result section.
2. Result section Page 9 line 171: "Fear and extinction memories are segregated in distinct IC projection neurons" Fig 2 (C, e-i). Please clarify if you used different animals for these experiments and how did you overlap labeled cells from different animals/experiments (Fear Ret vs. Ext.) (Fig 2c).
3. Page 19 line 382: The authors used slice physiology (Supp Fig 13) but slicing the brain would potentially affects these connections. Please add/discuss this potential limitation (and compare it with in-vivo single unit recording).

4. Page 20 line 399: Please specify the relevance of this result. For instance, IC neurons inhibit each other perhaps via interneurons.
5. Page 20 line 407: Please finish the paragraph suggesting the importance of this results.
6. The CPP or the CPA tasks/equipment's were not described in the methods.
7. In the discussion please do not repeat the results. Discuss more the implications and limitations/caveats of your results and methods. The current version is very general. For instance, talk about the role of other IC projections or afferents (i.e. PL, BLA, IL) be specific. Cite previous relevant studies. What would be the relevance of IC and its projections in other defensive behaviors (i.e. avoidance). Talk about similarities and differences between IC and OFC (be specific).
8. Figure 4. They mentioned chemogenetics but I did not more details about it (methods?).
9. Page 13 line 251: Fig. 4c is not properly labeled.
10. Fig 1 d, i: add the AP coordinates.
11. Fig 4 E: Please talk about the idea of stimulating IC-NAc during the Cond phase.
12. Fig 6a: The brain cartoon, please add an "OR" for the Retro AVV injection in CeA as well. Apply this to the other figures.
13. Supp Fig 2. Please show the histology (apply the same approach for the other opto experiments).
14. Supp Fig 3 d. Please add the legend.
15. Supp Fig 5 e-f. I could not find the ephys validation for the IC-NAc projection.
16. Supp Fig 10. The Fig is a bit confusing. Please show the group data for IC-CeA and IC-NAc and/or add more labels or information for the reader.
17. Supp Fig 12: Its interesting/surprising to see that the CP and NAc project to the IC neurons. Please discuss. Any leaky or limitation of the technique.
18. OFC inputs to IC neurons induce excitatory and inhibitory currents. Also suggesting polysynaptic connections. Please discuss/describe/clarify add limitations and caveats if any.
19. Be more specific with the title.
20. Report F values, T values.

Responses to Reviewers' comments (Reviewers' comments in *italic*)

Reviewer #1:

In this manuscript, Wang and colleagues investigate the roles of two distinct projection-defined insular cortex neuron subpopulations in fear conditioning and extinction. Major findings of this study include: i) Fear memory- and extinction memory-encoding neurons are found to be enriched in the anterior insular cortex (aIC) using the FosTRAP method. Furthermore, optogenetic inhibition of fear- and extinction-memory ensembles in aIC impaired memory retrieval for fear and extinction memories, respectively. ii) fear and extinction memories are segregated into distinct and non-overlapping IC projection neurons with central amygdala (CeA) and nucleus accumbens (NAc) projectors encoding fear and extinction memories, respectively. iii) IC-CeA and IC-NAc projectors reciprocally inhibit each other through intracortical interneurons. iv) whole-brain mapping of monosynaptic inputs to IC-CeA and IC-NAc projecting IC neurons reveal that IC-NAc projectors receive significantly more input from the orbitofrontal cortex (OFC). Furthermore, intercortical connectivity between OFC and IC projectors gate extinction learning and memory. v) Finally, activation of OFC to IC to NAc pathway promotes extinction learning and memory. The results of these experiments are convincing and provide important insights into the neural circuit mechanisms underlying fear and extinction memories.

Previous studies have demonstrated that IC-CeA and IC-NAc projection neurons, particularly ones located in the posterior IC, play an important role in aversive state processing (Gehrlach et al., Nature Neuroscience, 2019). Furthermore, a recent publication further showed that IC acts as a state-dependent regulator of fear which is necessary to set equilibrium between extinction and fear memories (Klein et al., Science, 2021). The present study complements and extends these previous findings significantly by investigating the role of projection-defined IC neurons in the anterior IC in fear and extinction learning. Overall, the present study is timely and makes an important contribution to the fear and extinction literature. However, there are points that need to be addressed.

We thank the reviewer for the positive comments.

Major points:

1. As mentioned above, in a paper published a few weeks ago Klein and colleagues demonstrate the role of IC in establishing the balance between fear and extinction memories. It is necessary to cite the Klein et al., Science, 2021 paper and discuss the findings of the present study in light of Klein et al. findings, particularly emphasizing the differences and the overlap between the two studies.

Cited and discussed accordingly (please see ref. 19; page 5, end of the 1st paragraph; page 23, end of the last paragraph and page 24).

2. There are several issues regarding the fiber photometry experiments in Figure 3:

i) The authors state that in calculating dF/F , F_0 refers to the median of fluorescence values during the baseline period which they took as the 5-s period before the onset of the first CS. However, the behavior of animals and their fear state changes with each CS through the course of fear conditioning and also the course of fear retrieval and extinction retrieval sessions. As a result, neuronal activity is expected to change through the course of these sessions as well. Therefore, 5-s period before the first CS would not reflect the calcium transients and fluorescence levels during the pre-CS period of each CS. Photobleaching during a session is also expected to affect baseline fluorescence levels through the course of these sessions. It is therefore more appropriate to use the baseline period for each CS (5-s before each CS) separately and analyze the dF/F for each CS by using the F_0 value calculated by using the 5-s period before each CS. Please re-analyze the dF/F values in Figure 3 accordingly.

We thank the reviewer for raising this error. In fact, the baseline period was defined as the 5-s preceding onset of each CS, rather than that of the first CS, which was misrepresented in the original version of the manuscript. This error has

been corrected in the revised manuscript (from page 34, middle of the 2nd paragraph).

ii) In Figure 3e and 3g, it would be good to see how the responses are after the tone offset. Please include at least 10-s post-tone period in these graphs. It would also be good to see a longer baseline period (e.g., 10-s) before the tone.

Done accordingly (please see new Fig. 3e, g and its legend).

iii) In Figure 3g, the fluorescence level during baseline changes from fear to extinction retrieval likely reflecting the change in fear state and movement level of the animals. In the heat maps, baseline activity seems to have high transients in most animals during extinction retrieval. For instance, in animal #1, there is increased activity during the CS however the activity is quite high before the CS, as well. It would therefore be good to see a longer baseline period before the CS, as I also stated in my previous comment. Also, since the variability in neural activity during baseline seems to be high in extinction retrieval, it would be good to report the data as z-scores which will take into consideration the standard deviation of the signal in baseline before each CS.

Following the reviewer's suggestion, we extended the baseline period to 10 s (please see our response to point #i). Moreover, we included the data of z-scores of calcium response in IC-CeA and IC-NAc projectors as follows (please see Fig. R1). Notably, these modified analyses verified the findings similar to the $\Delta F/F$ calculations, demonstrating that behaviorally related activity is specific to IC-CeA and IC-NAc projectors.

Fig. R1. Z-score analysis of response patterns of IC-CeA and IC-NAc projectors to fear learning and fear- or extinction-memory retrieval. IC-CeA projectors, $n = 7$ mice; IC-NAc projectors, $n = 9$ mice. **a, b** Z-scores of calcium signals recorded from IC-CeA projectors. **a** Left: Average calcium signals (z-score) of IC-CeA projectors aligned to the onset of the CS during fear conditioning. Thick lines, mean; shaded areas, S.E.M. Right: the peak and the area under the curves (AUC) during tone (in cyan box) and shock (in gray box). Tone peak: $F_{(1.940, 11.64)} = 3.159$, $P = 0.0813$; tone AUC: $F_{(1.236, 7.414)} = 3.485$, $P = 0.0977$; shock peak: $F_{(1.679, 10.07)} = 1.278$, $P = 0.3125$; shock AUC: $F_{(2.055, 12.33)} = 1.970$, $P = 0.1804$, one-way repeated-measure ANOVA. The IC-CeA projectors showed a tendency to increase in response to tones while the response to footshocks remained constant during fear learning. **b** Average calcium signals (z-score) of IC-CeA projectors aligned to the onset of the CS during fear- or extinction-memory retrieval. Left upper: calcium signals during fear memory retrieval, average calcium signals and heatmap of calcium signals in each mouse. Left lower: similar as above for calcium signals during extinction memory retrieval. Right: The peak and AUC during tone in fear- and extinction-memory retrieval (in cyan box). The activity of IC-CeA projectors was significantly higher during fear-memory retrieval than that during extinction-memory retrieval. Peak: $t_{(6)} = 5.051$, $*P = 0.0023$; AUC: $t_{(6)} = 73.504$, $*P =$

0.0128, paired Student's *t*-test. **c, d** Similar to **a, b** for calcium signals recorded from IC-NAc projectors. **c** The responses of IC-NAc projectors to both tone and foot shock kept constant during fear learning. Tone peak: $F_{(2.428, 19.42)} = 1.124$, $P = 0.3546$; tone AUC: $F_{(1.772, 14.17)} = 1.631$, $P = 0.2304$; shock peak: $F_{(2.335, 18.68)} = 1.244$, $P = 0.3160$; shock AUC: $F_{(1.944, 15.55)} = 1.237$, $P = 0.3161$, one-way repeated-measure ANOVA. **d** The activity of IC-NAc projectors was significantly lower during fear-memory retrieval than that during extinction-memory retrieval. Peak: $t_{(8)} = 3.809$, $^{**}P = 0.0052$; AUC: $t_{(8)} = 2.626$, $^{*}P = 0.0304$, paired Student's *t*-test.

iv) In Figures 3d and 3f, responses to the tone seem to increase particularly during the onset of the tone and these tone onset responses seem to increase for both IC-CeA and IC-NAc projectors through the course of fear conditioning. Analyzing the tone responses for the first 5-s (rather than the whole duration) of the CS might yield significant increases from the first to the last tone. How do the middle figures showing AUC values in Fig. 3d and 3f look if first 5-s of CS is analyzed? If there is a significant increase from first to last tone for both IC-CeA and IC-NAc projectors, the authors will need to reinterpret these findings, particularly for the IC-NAc projector group. Also, how do Figures 3e and 3g look if AUC is calculated for the first 5 sec of the tone?

Following the reviewer's suggestion, we re-analyzed the tone responses for the first 5-s of the CS (please see Fig. R2 below). Based on the newly analyzed results, the IC-CeA, but not the IC-NAc projector, showed a significant increase from the first to the last tone, during fear conditioning, which is consistent with the results of the previous analysis of the tone response for the entire duration. For memory retrieval shown in Fig. 3e, g, the lower response of IC-CeA projectors to extinction than fear memory retrieval largely remained with the analysis of the tone responses of the first 5-s of the CS. Moreover, by re-analyzing the tone responses of the first 5-s of the CS, the difference in IC-NAc projectors between extinction and fear memory retrieval also remained significantly. Based on the slow kinetics of GCaMP6m signal¹, we preferred to the criterion of analyzing the responses for the entire duration, as has been generally employed in the field².

References

1. Chen TW, Wardill TJ, Sun Y, Pulver SR, Renninger SL, Baohan A, Schreiter ER, Kerr RA, Orger MB, Jayaraman V, Looger LL, Svoboda K, Kim DS. Ultrasensitive fluorescent proteins for imaging neuronal activity. *Nature* **99**: 295-300 (2013).
2. Klein AS, Dolensek N, Weiland C, Gogolla N. Fear balance is maintained by bodily feedback to the insular cortex in mice. *Science* **374**: 1010-15 (2021).

Fig. R2. Quantification of response patterns of IC-CeA and IC-NAc projectors to fear learning and fear- or extinction-memory retrieval by analyzing the initial

responses to tone or shock. Only the first 5-s period of the CS and the 2-s period of the US were analyzed. IC-CeA projectors, $n = 7$ mice; IC-NAc projectors, $n = 7$ mice. **a, b** Calcium signals ($\Delta F/F$) recorded from IC-CeA projectors. **a** The peak and the area under the curves (AUC) during tone (upper) and shock (lower). Tone peak: $F_{(2.353, 14.12)} = 3.822$, $*P = 0.0417$; tone AUC: $F_{(2.317, 13.90)} = 5.837$, $*P = 0.0120$; shock peak: $F_{(1.182, 7.095)} = 0.8557$, $P = 0.4050$; shock AUC: $F_{(1.260, 7.559)} = 1.087$, $P = 0.3490$, one-way repeated-measure ANOVA. **b** Calcium signals during fear and extinction memory retrieval. Peak: $t_{(6)} = 2.365$, $P = 0.0559$; AUC: $t_{(6)} = 3.069$, $*P = 0.022$, paired Student's t -test. **c, d** Similar to **a, b** for calcium signals recorded from IC-NAc projectors. **c** Responses of IC-NAc projectors to both tone and foot shock kept constant during fear learning. Tone peak: $F_{(1.951, 15.61)} = 0.6899$, $P = 0.5129$; tone AUC: $F_{(2.108, 16.86)} = 0.6266$, $P = 0.5544$; shock peak: $F_{(2.213, 17.71)} = 1.207$, $P = 0.3266$; shock AUC: $F_{(2.065, 16.52)} = 0.6326$, $P = 0.5485$, one-way repeated-measure ANOVA. **d** Activity of IC-NAc projectors during fear and extinction memory retrieval. Peak: $t_{(8)} = 6.285$, $***P = 0.0002$; AUC: $t_{(8)} = 3.604$, $**P = 0.0069$, paired Student's t -test.

v) In Figures 3d and 3f, responses to shock are analyzed by calculating the AUC during the 20-s period (2-s shock and 18-s post-shock) after the tone. However, the shock is only 2-s long and it is not clear why the authors included such a long post-shock period into their analysis. To see responses to the shock, the authors should rather calculate the AUC values during the 2s shock period. How do the right figures in Figures 3d and 3f look if analysis is restricted to 2s?

Following the reviewer's suggestion, we restricted the analysis of shock response to 2 s (please see Fig. R2 above) and found constant responses of both IC-CeA and IC-NAc projectors to foot shocks. Again, considering the slow kinetics of GCaMP6m signal and the criterion generally employed in the field, we calculating the AUC during the 20-s period for the responses to shock is well rationalized.

3. In Figures 3, 4 and supplementary Figure 5, the freezing levels are quite different when the two projector groups are compared. For instance, in Figure 3c, mice in IC-CeA group freeze much less (~ 60%) compared to IC-NAc group (~80%) at the end of fear conditioning. Also, when we compare the control mice in Figure 4f versus 4g, the control mice in IC-CeA projector group (Fig. 4f) freeze significantly lower compared to the control mice in IC-NAc projector group (Fig. 4g). The fear conditioning

parameters seem to be same between these two groups, what can be the reason for this discrepancy in freezing levels between the groups? Please discuss the reasons for this discrepancy?

We thank the reviewer for raising the important issue of freezing data variability across the control groups when comparing the two projector groups. In fact, for the control mice in Fig. 4f versus 4g, different fear conditioning parameters were used. Indeed, optogenetic activation of IC-CeA projectors during fear conditioning with moderate (0.3 mA) foot shocks significantly enhanced the freezing behavior, and this enhancement was maintained in the fear-memory retrieval (Fig. 4f). By contrast, activation of IC-CeA projectors during fear conditioning with strong (0.5 mA) foot shocks failed to affect the freezing behavior (Supplementary Fig. 12b). To reveal the roles of IC-NAc projectors in the extinction memory, strong (0.5 mA) foot shocks were used during fear conditioning. Thus, different fear conditioning parameters largely account for the different freezing levels between the groups shown in the control mice in Fig. 4f versus 4g.

Furthermore, due to the importance of CeA in fear memory expression, injecting AAVs into CeA to manipulate the IC-CeA projectors would inevitably cause negligible damage to CeA, which resulted in lower levels of freezing in the IC-CeA group than in the IC-NAc group at the end of fear conditioning (please see Fig. 3c for an example). This behavioral variability should not challenge the reliability of the results as the control and experimental groups within the IC-CeA and IC-NAc groups were always compared and analyzed separately.

4. In Figure 1, there is inter-individual variability in freezing levels of mice with some mice freezing more and some less during fear retrieval (Fig. 1e) and extinction retrieval (Fig. 1j) tests. It would be interesting to see and also will strengthen the findings of this experiment if there is a correlation between freezing levels and the number of

FosTRAPed cells. For instance, one would expect that the more number of FosTRAPed cells in an animal will correlate with higher freezing levels during fear retrieval and vice versa for extinction retrieval.

Done accordingly. However, the present results do not indicate a clear association between freezing levels and the number of FosTRAPed cells during fear or extinction memories (please see Fig. R3 below), implicating that the size of the IC neuronal ensemble is not the only factor in deciphering the expression of fear or extinction. Other factors such as neuronal projection, are probably also important.

Fig. R3. Correlation of FosTRAPed IC fear- and extinction-memory ensembles and freezing level. Pearson coefficient (R) and P values were indicated for each plot.

5. In Figures 3d and 3f (middle and right), the authors state that they performed one-way ANOVA. Did they perform repeated measures? If not, repeated measures ANOVA should be done.

Following the reviewer's suggestion, we performed repeated measures ANOVA in the revised manuscript (please see new Fig. 3d, f, and its legend).

6. In Supplementary Figure 13, the authors find that OFC inputs to IC-CeA projectors provided more inhibition than excitation indicating a net effect of inhibition of this input on these IC neurons. Therefore, one would expect that if OFC input to IC is activated during fear conditioning, this should lead to impaired fear learning and retrieval since IC-CeA neurons are inhibited. However, optogenetic activation of OFC input to IC in Figure 6e does not have any effect on fear acquisition and retrieval contrary to expectation. Can the authors discuss possible reasons for this.

Discussed accordingly (please see page 21, beginning of the 1st paragraph).

Minor points:

7. The authors should include histological verification of optic fiber placements and virus expression profiles of all mice in each of the experiments as supplementary figures.

Done accordingly (please see new Supplementary Figs. 2e–g, 5f, 7, 8h, i, 9c, d, 11d, e, 21 and their legends).

Reviewer #2:

In this study, the authors characterized how the insular cortex (IC) mediates fear expression and extinction through projections to the central amygdala (CeA) and the nucleus accumbens (NAc), respectively. Using state-of-the-art engram tagging, optogenetics, chemogenetics, viral intersectional approaches, photometry, slice electrophysiology and tracing, the authors found how IC-CeA and IC-NAc circuits were necessary and sufficient for fear and extinction memories, respectively. Moreover, the authors found that IC-CeA and IC-NAc neurons gated each other's activity through local feedforward inhibition, and that the orbitofrontal cortex promoted fear extinction

through projections to the IC.

This is an outstanding, innovative and creative study that will have a positive impact in the Pavlovian fear field. The approach was straightforward, the experiments were comprehensive and the narrative was clear and concise. I have Identified several issues that I hope the authors can address before moving forward with this manuscript.

We thank the reviewer for the positive comments.

Major issues:

1. The study lacks histology reports throughout, aside from single subject examples. The authors need to report virus spread, and optic fiber and placements for all corresponding experiments.

Done accordingly (please see new Supplementary Figs. 2e–g, 5f, 7, 8h, i, 9c, d, 11d, e, 21 and their legends).

2. The authors should report how their optogenetic and chemogenetic, or ablation manipulations affect general locomotion.

Done accordingly (please see new Supplementary Figs. 7c, d, 10, 14 and their legends).

Minor issues:

3. The authors should show the fear conditioning freezing curves in Figure 1. It is not clear why the authors tagged neurons after two extinction sessions, as this would include neurons involved in fear retrieval (early extinction session), extinction retrieval (because they already extinction the previous day) and further extinction learning. Reporting the freezing curves will allow the reader to better interpret the results in Figure 1.

Done accordingly (please see new Supplementary Fig. 1a, c, and its legend; page 7, middle of the 1st paragraph, and page 8, middle of the 1st paragraph).

4. In Figure 1, when the authors report extinction cell tagging, they reported a CS only control, but not a fear reactivation control group. This is important because fear extinction retrieval involves both fear and extinction retrievals. It is unclear if the tagged cells account for fear or extinction reactivation because we do not know how many cells would have been recruited by fear retrieval on its own.

Done accordingly (please see new Fig. 1h–l, and its legend; page 8, end of the 1st paragraph, and page 9, beginning of the 1st paragraph).

5. It would be good to see the mean fluorescence of IC-CeA and IC-NAC ensembles outside of the fluorescence ratio as well.

Done accordingly (please see new Supplementary Fig. 3c, d and its legend; page 10, middle of the last paragraph).

6. Photometry experiments lack a within subject control (mCherry or isosbestic channel) to account for movement artifacts.

Done accordingly (please see new Supplementary Fig. 6, and its legend; page 12, end of the 1st paragraph).

7. The authors should be cautious when describing effects on fear or extinction retrieval after employing manipulations that affected fear or extinction acquisition. To clearly see effects on expression, the authors need to limit their manipulations to the expression test period and avoid confounds of acquisition effects.

Done accordingly (please see new Supplementary Fig. 9 and its legend; page 13,

end of the last paragraph to page 14, beginning of the 1st paragraph).

8. *The authors claim that there was a significant increase in CPP score in in Line 281, but Figure 8u depicts a p value of 0.09, which did not reach significance.*

Corrected accordingly (please see page 15, end of the 1st paragraph).

9. *What is the rationale of having different tone durations for conditioning, extinction and retrieval tests?*

Rephrased (please see page 29, middle of the 1st paragraph).

10. *The authors claim that activation of OFC-IC projections enhanced fear extinction without affecting fear conditioning (Figure 6e). We know from Extended Data Figure 7b that stimulation of IC-CeA does not further enhance fear conditioning when using a high shock. Thus, it is possible that using a high shock in Figure 6e could have mask an effect of fear enhancement from stimulating OFC-IC projections.*

**Discussed accordingly (please see page 21, beginning of the 1st paragraph).
OFC→IC inputs more strongly activated IC-NAc projectors than IC-CeA projectors. The finding that activation of OFC→IC inputs during the fear conditioning did not affect the formation and expression of fear memory is similar to the effects induced by activation of IC-NAc projectors (please see new Supplementary Fig. 12 and its legend).**

11. *The authors report that 100% of IC-NAc projectors receive direct excitatory inputs from OFC, whereas IC-CeA only received 69%, suggesting that OFC projections to IC are biased towards NAc projectors (Extended Data Figure 13c). It is unclear whether the authors can conclude this with only 18 and 16 neurons per projection. The authors should employ a statistical test, such as a Pearson's, to corroborate their claim.*

Corrected accordingly. We performed a non-parametric Fisher's exact test, which suggested the statistical significance supporting more abundant OFC inputs to IC-NAc projectors (please see Supplementary Fig. 19d and its legend; page 20, middle of the 1st paragraph).

12. Is Extended Data Figure 10a group data? If so, the authors should report the error bars.

No, the original Supplementary Fig. 10a (now Supplementary Fig. 17a) is not group data. Instead, it is just a representative example of synaptic connection from IC-CeA projector to IC-NAc projector. In response to the reviewer's concern, the group data on synaptic latency are included in the revised manuscript (please see new Supplementary Fig. 17c, d and its legend).

Reviewer #3:

In the present study, the authors made an excellent and complete job on studying the role of neuronal projections coming from the Insular cortex (IC) to either the Central Amygdala (CeA) or to the Nucleus Accumbens (NAc) during fear conditioning and extinction phases (learning and memory). The authors show that while IC-CeA projection is critical for fear conditioning (and its retrieval/expression), the IC-NAc projection is important for extinction learning and its retrieval. Interestingly, they also showed that when one of these projections is active, the other projection is indirectly inhibited (reciprocal inhibition between IC neurons projecting to either CeA or to NAc, perhaps via interneurons). Lastly, the authors dissected the afferent inputs coming from different brain regions to IC neurons projecting to CeA or to NAc. Then, they subsequently focused on the OFC contribution in regulating fear conditioning and extinction phases. The authors particularly showed that activating the OFC-IC-NAc pathway facilitated extinction learning and its memory (retrieval). Overall, this is a

very compelling and important set of findings that complement previous data in the field. I do have several suggestions/comments to be addressed in a revision.

We thank the reviewer for the positive comments.

Comments:

1. The results section is hard to follow (disorganized). Please check the sequence/flow of the main and supplementary figures in the result section.

Many thanks to the reviewer's kind suggestion. We have double checked the sequence/flow of the main and supplementary figures in the result section and made corrections.

2. Result section Page 9 line 171: "Fear and extinction memories are segregated in distinct IC projection neurons" Fig. 2 (C, e-i). Please clarify if you used different animals for these experiments and how did you overlap labeled cells from different animals/experiments (Fear Ret vs. Ext.) (Fig. 2c).

We indeed used different animals for these experiments. In response to the reviewer's concern, we segregated these two groups for clarification (please see new Fig. 2c and its legend).

3. Page 19 line 382: The authors used slice physiology (Supp Fig 13) but slicing the brain would potentially affects these connections. Please add/discuss this potential limitation (and compare it with in-vivo single unit recording).

The potential limitation was added to the revised manuscript (please see page 26, end of the 1st paragraph).

4. Page 20 line 399: Please specify the relevance of this result. For instance, IC neurons

inhibit each other perhaps via interneurons.

Rephrased (please see page 20, end of the 1st paragraph).

5. Page 20 line 407: Please finish the paragraph suggesting the importance of this results.

Rephrased (please see page 21, end of the 1st paragraph).

6. The CPP or the CPA tasks/equipment's were not described in the methods.

Included accordingly (please see page 36, the 1st paragraph to page 38, the 1st paragraph).

7. In the discussion please do not repeat the results. Discuss more the implications and limitations/caveats of your results and methods. The current version is very general. For instance, talk about the role of other IC projections or afferents (i.e. PL, BLA, IL) be specific. Cite previous relevant studies. What would be the relevance of IC and its projections in other defensive behaviors (i.e. avoidance). Talk about similarities and differences between IC and OFC (be specific).

Rephrased (please see the new Discussion section).

8. Figure 4. They mentioned chemogenetics but I did not more details about it (methods?).

We apologize for the inaccurate description. In the previous version of our manuscript, we inappropriately referred to the use of light chain of tetanus toxin (TetTox) to inactivate the synaptic release of IC projectors as “chemogenetics”. We rephrased this statement in the revised manuscript (please see new Fig. 4 and

its legend; page 12, end of the last paragraph, and page 13, beginning of the 1st paragraph).

9. Page 13 line 251: Fig. 4c is not properly labeled.

Corrected accordingly (please see page 14, beginning of the 2nd paragraph).

10. Fig 1 d, i: add the AP coordinates.

Corrected accordingly (please see new Fig. 1d, i and its legend).

11. Fig 4e: Please talk about the idea of stimulating IC-NAc during the Cond phase.

Done accordingly (please see new Supplementary Fig. 13 and page 14, middle of the 2nd paragraph).

12. Fig 6a: The brain cartoon, please add an “OR” for the Retro AVV injection in CeA as well. Apply this to the other figures.

Done accordingly (please see new Figs. 3a, 4a, e, 5a, 6a, and Supplementary Figs. 15a, 20a and their legends).

13. Supp Fig 2. Please show the histology (apply the same approach for the other opto experiments).

Done accordingly (please see new Supplementary Fig. 2e and its legend).

14. Supp Fig 3 d. Please add the legend.

Done accordingly (please see the legend for new Supplementary Fig. 4d).

15. *Supp Fig 5 e-f. I could not find the ephys validation for the IC-NAc projection.*

Done accordingly (please see new Supplementary Fig. 8c and its legend).

16. *Supp Fig 10. The Fig is a bit confusing. Please show the group data for IC-CeA and IC-NAc and/or add more labels or information for the reader.*

Done accordingly (please see new Supplementary Fig. 17 and its legend).

17. *Supp Fig 12: Its interesting/surprising to see that the CP and NAc project to the IC neurons. Please discuss. Any leaky or limitation of the technique.*

We thank the reviewer for raising this important issue. After careful re-examination of the original data, we noticed that an error was introduced in the previous version of our manuscript. We incorrectly identified a portion of the IC area as the caudoputamen (CP) and a portion of the piriform (PIC) area as NAc. We have corrected this error in the revised manuscript (please see new Supplementary Fig. 19 and its legend).

18. *OFC inputs to IC neurons induce excitatory and inhibitory currents. Also suggesting polysynaptic connections. Please discuss/describe/clarify add limitations and caveats if any.*

Rephrased (please see the new Supplementary Fig. 20e and its legend; and page 20, middle of the 1st paragraph).

19. *Be more specific with the title.*

Rephrased (please see the new Title).

20. Report *F* values, *T* values.

Done accordingly (please see the new Figure legends throughout the whole manuscript).

REVIEWERS' COMMENTS

Reviewer #1 (Remarks to the Author):

The authors have responded to the issues I have raised in my previous review adequately. I do not have any further comments and recommend the manuscript for publication.

Reviewer #2 (Remarks to the Author):

The authors addresses all of my concerns. This is an outstanding study.

Reviewer #3 (Remarks to the Author):

All my comments were appropriately addressed. This version of the manuscript is much better.

Responses to Reviewers' comments (Reviewers' comments in *italic*)

Reviewer #1:

The authors have responded to the issues I have raised in my previous review adequately. I do not have any further comments and recommend the manuscript for publication.

We thank the reviewer for the positive comments.

Reviewer #2:

The authors address all of my concerns. This is an outstanding study.

We thank the reviewer for the positive comments.

Reviewer #3:

All my comments were appropriately addressed. This version of the manuscript is much better.

We thank the reviewer for the positive comments.